# A single bout of prior resistance exercise attenuates muscle atrophy and declines in myofibrillar protein synthesis during bed-rest in older men

Benoit Smeuninx[1,2] , Yasir S. Elhassan[3,4], Elizabeth Sapey[5], Alison B. Rushton[1], Paul T. Morgan[1], Marie Korzepa[1], Archie E. Belfield[1], Andrew Philp[6], Matthew S. Brook[7,8] , Nima Gharahdaghi[7,8], Daniel Wilkinson[7,8] , Kenneth Smith[7,8] , Philip J. Atherton[7,8] and Leigh Breen[1,9,10]

[1]*School of Sport, Exercise and Rehabilitation Sciences, University of Birmingham, Birmingham, UK*
[2]*Cellular & Molecular Metabolism Laboratory, Monash University, Melbourne, Victoria, Australia*
[3]*Institute of Metabolism and Systems Research, University of Birmingham, Birmingham, UK*
[4]*Centre for Endocrinology, Diabetes and Metabolism, Birmingham Health Partners, Birmingham, UK*
[5]*Birmingham Acute Care Research Group, Institute of Inflammation and Ageing, University of Birmingham, Birmingham, UK*
[6]*Centre for Healthy Ageing, Centenary Institute, Camperdown, New South Wales, Australia*
[7]*Centre Of Metabolism, Ageing and Physiology (COMAP), School of Medicine, University of Nottingham, Royal Derby Hospital Centre, Derby, UK*
[8]*MRC-ARUK Centre of Excellence for Musculoskeletal Ageing Research, University of Nottingham, Derby, UK*
[9]*MRC-ARUK Centre for Musculoskeletal Ageing Research, University of Birmingham, Birmingham, UK*
[10]*NIHR Birmingham Biomedical Research Centre, University Hospitals Birmingham NHS Foundation Trust and University of Birmingham, Birmingham, UK*

Handling Editors: Karyn Hamilton & Christopher Sundberg

The peer review history is available in the Supporting Information section of this article (https://doi.org/10.1113/JP285130#support-information-section).

**Abstract**  Impairments in myofibrillar protein synthesis (MyoPS) during bed rest accelerate skeletal muscle loss in older adults, increasing the risk of adverse secondary health outcomes. We investigated the effect of prior resistance exercise (RE) on MyoPS and muscle morphology during a disuse event

The Journal of Physiology

in 10 healthy older men (65–80 years). Participants completed a single bout of unilateral leg RE the evening prior to 5 days of in-patient bed-rest. Quadriceps cross-sectional area (CSA) was determined prior to and following bed-rest. Serial muscle biopsies and dual stable isotope tracers were used to determine rates of integrated MyoPS (iMyoPS) over a 7 day habitual 'free-living' phase and the bed-rest phase, and rates of acute postabsorptive and postprandial MyoPS (aMyoPS) at the end of bed rest. Quadriceps CSA at 40%, 60% and 80% of muscle length significantly decreased in exercised (EX) and non-exercised control (CTL) legs with bed-rest. The decline in quadriceps CSA at 40% and 60% of muscle length was attenuated in EX compared with CTL. During bed-rest, iMyoPS rates decreased from habitual values in CTL, but not EX, and were significantly different between legs. Postprandial aMyoPS rates increased above postabsorptive values in EX only. The change in iMyoPS over bed-rest correlated with the change in quadriceps CSA in CTL, but not EX. A single bout of RE attenuated the decline in iMyoPS rates and quadriceps atrophy with 5 days of bed-rest in older men. Further work is required to understand the functional and clinical implications of prior RE in older patient populations.

(Received 14 July 2023; accepted after revision 5 October 2023; first published online 20 October 2023)
**Corresponding author** L. Breen: School of Sport, Exercise and Rehabilitation Sciences, University of Birmingham, Birmingham, B15 2TT, UK.     Email: L.breen@bham.ac.uk

**Abstract figure legend** Healthy older men performed a single bout of unilateral resistance exercise the evening prior to 5 days of in-patient bed-rest. Bed-rest resulted in significant declines in physical activity, dietary quality and adverse changes in blood metabolic health markers. Quadriceps muscle mass decreased following bed-rest in the exercised (EX) and non-exercised control leg (CTL) but was attenuated in the EX leg. Myofibrillar protein synthesis rates declined over the course of bed-rest compared with habitual pre-bed-rest rates in the CTL leg but were unchanged in the EX leg. Similarly, an expected postprandial increase in acute myofibrillar protein synthesis was observed in the EX leg following bed-rest, whereas no postprandial increase was detected in the CTL leg. The boldness and direction of arrows indicates the significance of the change.

## Key points

- Age-related skeletal muscle deterioration, linked to numerous adverse health outcomes, is driven by impairments in muscle protein synthesis that are accelerated during periods of disuse.
- Resistance exercise can stimulate muscle protein synthesis over several days of recovery and therefore could counteract impairments in this process that occur in the early phase of disuse.
- In the present study, we demonstrate that the decline in myofibrillar protein synthesis and muscle atrophy over 5 days of bed-rest in older men was attenuated by a single bout of unilateral resistance exercise performed the evening prior to bed-rest.
- These findings suggest that concise resistance exercise intervention holds the potential to support muscle mass retention in older individuals during short-term disuse, with implications for delaying sarcopenia progression in ageing populations.

**Benoit Smeuninx** obtained a bachelor's and master's degree in Sport Sciences at the KU Leuven (Belgium). He then completed a second master's degree in Exercise Physiology at Loughborough University (UK) before completing his PhD and postdoctoral studies at the University of Birmingham (UK) under the supervision of Professor Leigh Breen. During his time at the University of Birmingham, Benoit investigated mechanisms that lead to, but also prevent, age-related muscle mass loss. Currently, Benoit is investigating the molecular mechanisms that underpin the transition from non-alcoholic steatohepatitis to hepatocellular carcinoma under the supervision of Professor Mark Febbraio at Monash University (Australia).

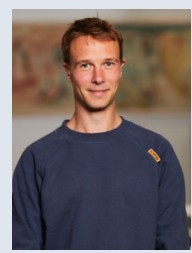

## Introduction

Skeletal muscle mass and strength loss (sarcopenia) is associated with an increased risk of frailty, falls, metabolic disease and all-cause mortality in older individuals (Lee et al., 2016; Veronese et al., 2019). It is estimated that as many as 32 million older individuals across Europe could be diagnosed with sarcopenia by 2045 (Ethgen et al., 2017). Alarmingly, the annual cost of hospitalization in those with sarcopenia was reported to be $40.9 billion in the USA in 2014, or ∼4% of the total national healthcare expenditure budget (Goates et al., 2019), whilst healthcare costs are ∼2–3 times greater in those with muscle weakness (Steffl et al., 2017). Thus, sarcopenia poses a major current and future predicted threat to healthcare resources and societies globally.

Skeletal muscle proteostasis is dependent on the equilibrium between muscle protein synthesis and breakdown. In older adults, the postprandial myofibrillar protein synthesis (MyoPS) response to amino acid nutrition is diminished (anabolic resistance) (Cuthbertson et al., 2005; Moore et al., 2015), and probably underpins age-related sarcopenia progression (Breen & Phillips, 2011). Periods of disuse and inactivity, typical during illness and hospitalization, induce rapid muscle anabolic resistance and atrophy in older individuals (Breen et al., 2013; Tanner et al., 2015). Older individuals often do not fully recover pre-admission levels of muscle mass, strength and function following a period of disuse compared with the young (Loyd et al., 2018; Rejc et al., 2018; Suetta et al., 2009). Furthermore, low muscle mass/attenuation and poor physical function at discharge are associated with a greater risk of readmission, longer length of stay (LoS) and reliance on external care after discharge (Covinsky et al., 1997; Fortinsky et al., 1999; Prado et al., 2018). Hence, disuse initiates a vicious cycle of accelerated muscle deterioration and heightened risk of disability and disease in older adults (Rezus et al., 2020). Therefore, mitigating disuse-induced muscle atrophy could delay sarcopenia progression, reduce associated healthcare expenditure and support quality of life in ageing populations.

Resistance exercise (RE) increases the utilization of amino acids from dietary protein for MyoPS in older adults (Pennings et al., 2011; Shad et al., 2016; Yang et al., 2012), a response that may persist for up to 72 h of post-exercise recovery (Burd et al., 2011; McKendry et al., 2019; Miller et al., 2005; Phillips et al., 1997). This is noteworthy as impairments in muscle protein stasis (particularly MyoPS) and the rate of muscle atrophy are most pronounced in the initial days of bed-rest (Di Girolamo et al., 2021; Hardy et al., 2022; Marusic et al., 2021; Tanner et al., 2015). It was recently demonstrated that a prior bout of damaging eccentric exercise increased MyoPS and prevented quadriceps muscle atrophy over the first 2 days of limb immobilization in younger adults (Jameson et al., 2021). Thus, it is possible that prior RE could offset the decline in MyoPS and muscle mass during short-term bed-rest in older adults, with relevance for patients undergoing planned elective surgery, for whom the typical LoS after hospital admission is 3–6 days (Papalia et al., 2021; Zhu et al., 2017). Although multi-modal exercise prehabilitation in the weeks prior to disuse is necessary to optimize physiological conditioning/reserve and improve clinically relevant outcomes in older patients (Carli & Zavorsky, 2005), concise RE prior to bed-rest offers a potentially complementary, cost-effective approach to support muscle mass during bed-rest, particularly when longer-term prehabilitation is not possible (e.g. stand-by surgery, inability/limited access to regular exercise) and warrants investigation.

In the present study, we determined the effects of a single bout of prior unilateral leg RE on quadriceps muscle morphology with 5 days of bed-rest in older individuals. Using dual-stable isotope tracers and serial muscle biopsy sampling, we determined integrated MyoPS (iMyoPS) rates over a consecutive 7 day habitual phase and 5 days of bed-rest, and acute postabsorptive and postprandial myofibrillar protein synthesis rates (aMyoPS) at the end of bed-rest. The protein and gene expression of key regulatory signalling intermediates was also investigated. We hypothesized that the decline in muscle mass and iMyoPS during bed-rest would be attenuated by prior RE compared with the untrained control leg, along with greater/preserved postprandial aMyoPS stimulation and anabolic signalling expression at the end of bed-rest.

## Methods

### Participants

Ten healthy older men (65–80 years) were recruited through local advertisements and deemed eligible for study participation if they had no history of structured RE training within 10 years prior to study participation, were deemed healthy and free of sarcopenia diagnosis as assessed by a general health questionnaire, a score of ≥9 on the Short Physical Performance Battery (SPPB) test, appendicular lean mass of ≥7.0 kg m$^{-2}$ and a body mass index (BMI) <30 kg m$^{-2}$. Participants were excluded from study participation if they had a coagulation disorder, myocardial infarction, artery/vein disease, chronic/systemic illness or had undergone hormone replacement therapy. Furthermore, participants were excluded if they currently smoked and consumed any anticoagulant medication or medication that might affect muscle metabolism. All participants were informed of the study purpose and procedures and provided their written informed consent. Ethical approval was obtained through the West Midlands Black Country Research Ethics Committee (16/WM/0483) and was registered at

clinicaltrials.gov (NCT04422665; RG_16-100). The study conformed to the standards outlined by the Declaration of Helsinki (7th edn). The present data set was collected as part of a larger trial, the findings of which have been published elsewhere (Smeuninx et al., 2021). All data presented herein are original and have not been published elsewhere.

## Experimental design

After an initial screening visit and obtainment of study consent, participants visited the Wellcome Trust/NIHR Clinical Research Facilities (CRF) at the Queen Elizabeth Hospital Birmingham for a preliminary testing visit (Day 1), resistance exercise session (Day 7), pre-bed-rest testing (Day 8), bed-rest phase (Days 8–13) and post-bed-rest testing day (Day 13). For each testing visit, participants reported to the CRF at 08.00 h after an overnight fast (or were already present in the case of Day 13). An overview of the study timeline is depicted in Fig. 1.

**Preliminary testing visit (Day 1).** Following an overnight fast, participants provided a single saliva sample, and a fasted venous blood sample was obtained, after which a baseline muscle biopsy from the vastus lateralis was obtained under 1% lidocaine using the Bergström needle technique, as described in our previous work (Bergstrom, 1975; Smeuninx et al., 2017). Participant height, body

mass, compartmental body composition, estimated 1 repetition maximum (1RM) and physical function were then determined (described below). Participants were then provided with a bolus of deuterated water ($D_2O$) and daily top-up doses (described below) for the measurement of iMyoPS. Prior to leaving the CRF, participants were given a 3 day weighed food diary to be completed during the subsequent prehabilitation phase, to determine habitual dietary intake. Participants were fitted with a hip-worn pedometer and wrist-worn accelerometer for the remainder of the study to monitor physical activity levels/intensity throughout prehabilitation and bed-rest phases.

**Resistance exercise session (Day 7).** Participants reported to the CRF between 15.00 and 18.00 h to complete a bout of unilateral RE prior to beginning 5 days of bed-rest the following morning. RE was performed by the strongest leg, as determined by the 1RM assessment during the preliminary testing visit. Each RE bout consisted of two warm-up sets at 50% of 1RM, followed by six sets at 75% of 1RM for both the leg extension and leg curl. RE sets consisted of a target 12 repetitions and were separated by 2 min of passive rest. The exercise load was adjusted to maintain a subjective rating of perceived exertion of 8–9 on the modified Borg category-ratio scale (CR-10) (Buckley & Borg, 2011).

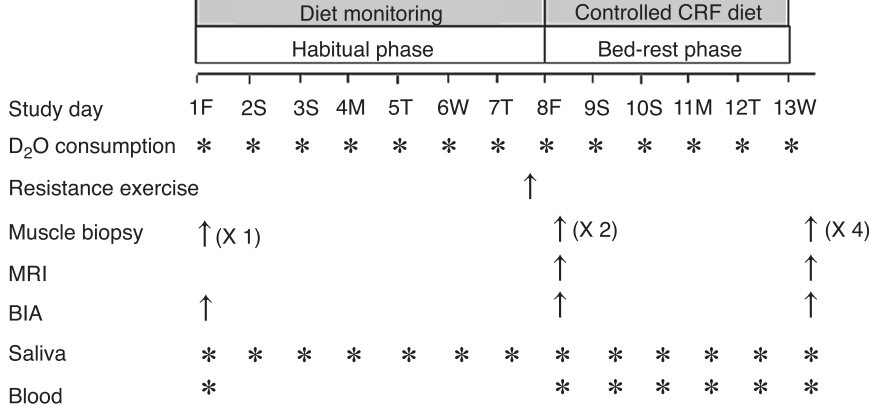

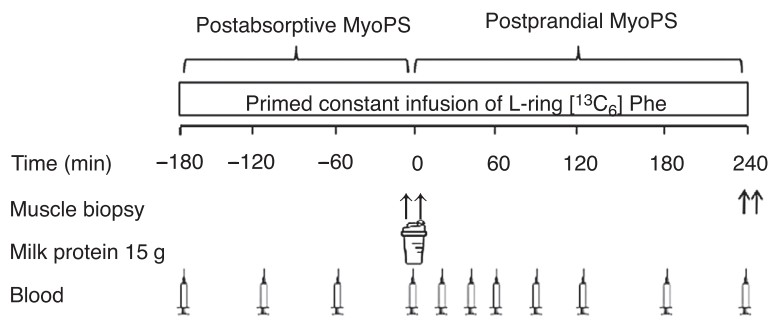

**Figure 1. Schematic overview of experimental design**
Schematic overview of the longitudinal experimental design (Days 1–13; top) and the acute stable isotope infusion trial conducted at the end of the bed-rest period (Day 13; bottom).

**Pre-bed rest testing visit (Day 8).** On the morning of Day 8 and following an overnight fast, participants reported to the CRF. A skeletal muscle biopsy from the vastus lateralis was obtained from both the exercised (EX) and non-exercised control leg (CTL). Following this, repeat assessments of height, body mass and body composition were taken. Finally, participants underwent a magnetic resonance imaging (MRI) scan to determine quadriceps muscle cross-sectional area (CSA) in both legs (described below).

**Bed-rest phase (Days 8–13).** To mimic the effects of a traditional inpatient hospital stay, participants underwent a 5 day period of strict bed-rest. Once participants returned from the mid-testing MRI scan, the bed-rest period commenced, and participants remained in bed. During the day, participants were allowed to sit up in bed or in a recliner chair. Bathing and sanitary activities were performed in a wheelchair. Accelerometer and pedometer devices were briefly removed during showering. To prevent coagulation disorders and bed sores, participants were given light, non-weight-bearing exercises to be performed hourly each day (e.g. knee bends, rolling side to side, ankle rotations). Participants also wore lower limb compression stockings and received a single daily subcutaneous enoxaparin injection (20 mg), at the same time each day (08.00–09.00 h), to prevent venous thromboembolism. Blood samples were obtained daily during bed-rest for analysis at the University Hospital Birmingham Clinical Laboratory for markers of coagulation (D-dimers, pro-thrombin). No adverse events relating to venous thromboembolism were reported. To mimic a typical in-patient stay, participants chose from a selection of meals/snacks provided by the CRF during bed-rest. Dietary energy and macronutrient intake was not strictly controlled but was closely monitored and logged by CRF nursing staff.

**Post-bed rest experimental trial (Day 13).** Following an overnight fast, participants were woken at 06.00 h. A 21G cannula was inserted in an antecubital vein of both forearms. One cannula was used for serial blood sampling whilst the other one was used to administer a stable amino acid isotope infusion. After obtaining a baseline blood sample, a primed-continuous infusion of L-[ring-$^{13}C_6$] phenylalanine was initiated (prime: 2 $\mu$mol kg$^{-1}$; infusion: 0.05 $\mu$mol kg$^{-1}$, Cambridge Isotope Laboratories, Andover, MA, USA). Blood samples were drawn from the contralateral arm at $-185$, $-120$, $-60$ and $-5$ min prior to and 20, 30, 60, 90, 120, 180 and 240 min after consumption of a milk protein drink (described below). Blood samples were collected in serum separator and EDTA-treated vacutainers (BD Biosciences, Oxford, UK) and centrifuged at 3000 r.p.m. at 4°C, with serum and plasma aliquots stored at $-80$°C

for further analyses. After 150 min of infusion, a muscle biopsy was obtained from the vastus lateralis of both legs. Immediately after biopsy obtainment, participants consumed 18.75 g of milk protein isolate (Myprotein, Cheshire, UK), providing 15 g of protein, dissolved in 300 mL of water. The amino acid composition of the milk protein is described in our previous work (Smeuninx et al., 2017). At 240 min after drink consumption, a second muscle biopsy from the vastus lateralis of each leg was obtained ∼3 cm proximal to the first biopsy and indicated the end of the infusion trial. Participants were then fed a meal of their choice and transported in a wheelchair to the MRI scanner for post-bed-rest measurement of quadriceps muscle CSA. Obtainment of the MRI scan indicated the end of the bed-rest phase. Participants walked back to the CRF for a final body composition assessment and consultation/assessment with a trained physiotherapist prior to discharge.

## Experimental procedures

**Body mass, height and body composition.** Participants' body mass and height were determined in light clothing to the nearest 0.1 kg and 0.1 cm using electronic weighing scales and a stadiometer, respectively. Compartmental body composition was determined using bioelectrical impedance analysis (TANITA BC-148), with participants holding an electrode in each hand whilst standing barefoot on two other electrodes. Participants were asked to consume 0.5 L of water 30 min before bioelectrical impedance assessments to standardize hydration status, having refrained from fluid consumption beforehand.

**Maximal strength assessment.** Participant knee extensor and flexor estimated 1RM strength was assessed for both legs separately. The leg reaching the highest estimated 1RM was assigned to the exercise intervention. Leg extension 1RM for both right and left leg was determined first, after which the protocol was repeated to determine leg flexor 1RM. Briefly, participants performed a one-set warm-up consisting of 12 repetitions at 10 kg. Thereafter, exercise load was gradually increased over subsequent sets until participants were unable to perform >10 repetitions. Increments in exercise load were based on subjective ratings of perceived exertion using the modified Borg category-ratio scale (CR-10) (Buckley & Borg, 2011). The Brzycki equation was used to estimate 1RM for both knee extensor and flexor strength.

**D2O dosing protocol.** The D$_2$O dosing protocol consisted of a loading day and 12 maintenance days. On day 1 of the trial and after providing a baseline saliva sample, participants consumed a loading dose of 70% D$_2$O equalling three times their body mass in millilitres with the aim to label the body water pool to ∼0.3 atom per cent excess (APE). The loading dose was split up

in 50 mL doses and consumed every 30 min to avoid nausea and light-headedness. Body water enrichment was maintained in a pseudo steady state using daily top ups, based upon a 6% per day decay rate (Wilkinson et al., 2014). Participants were instructed to provide a daily saliva sample upon waking, followed by consumption of the $D_2O$ maintenance dose. The $D_2O$ loading and maintenance protocol was well tolerated, and no adverse effects were reported by the participants.

**Quadriceps cross-sectional area.** Quadriceps CSA was determined on Days 8 and 13 of the experimental trial using a 3T MRI scanner (Phillips Achieva 3T scanner) at the Birmingham University Imaging Centre (BUIC). Participants were placed on the scanner bed in a supine position and entered the scanner feet first. To ensure consistent positioning across pre- and post-bed-rest MRI scans and participant comfort, participants' feet were taped to internally rotate the toes, and sandbags placed over the ankles. To aid alignment of pre- and post-bed-rest MRI scans, cod liver oil tablets were taped every 5 cm on the lateral part of the limb starting at the fibula head running proximal up to the greater trochanter. Accurate placement of cod liver oil capsules was achieved by re-marking the sites of capsule placement throughout the bed-rest phase using a non-toxic pen. Images of both thighs in the axial plane were obtained starting at the top of the patella covering the full length of the femur. Four stacks of five images, interspaced by 5 mm, were taken totalling 25 axial images. Quadriceps femoris CSA of both legs was determined at 20%, 40%, 60% and 80% of the length between the top of the patella and the greater trochanter, with the measurer blinded to leg order, using standard imaging software (OsiriX medical imaging software, OsiriX, Atlanta, GA, USA). Intra-individual reliability for quadriceps CSA calculated from the repeated analysis of three MRI scans at 20%, 40%, 60% and 80% of limb length was 1.4%, 1.0%, 0.8% and 0.9%, respectively. Within-participant coefficient of variation for repeat day-to-day quadriceps CSA measurements was 1.1%.

### Sample analyses

**Plasma amino acids, plasma isotope enrichment and body water $^2H$ enrichment.** Plasma $[^{13}C_6]$ phenylalanine enrichment was determined by GCMS (model 5973; Hewlett Packard, Palo Alto, CA, USA) by monitoring ion 234/240. Briefly, 100 $\mu$L of plasma was diluted 2:1 with acetic acid before being purified through cation-exchange columns and eluted using 2 M $NH_4OH$. Eluents were dried down under nitrogen and converted to their N-tert-butyldimethyl-silyl-N-methyltrifluoracetamide (MTBST-FA) derivative. Leucine and phenylalanine concentrations were measured using internal standards U-$[^{13}C_6]$ leucine (ions 302/308) and U-$[^{13}C_9-^{15}N]$ phenylalanine (ions

336/346). Body water enrichment was measured as described by Wilkinson et al. (2014). Briefly, 50 $\mu$L of saliva was heated in an inverted auto-sampler vial for 4 h at 100°C and placed upright on ice afterwards to condense extracted body water. This was then transferred to a clean auto-sampler vial and a total of 0.1 $\mu$L of body water was injected into a high-temperature elemental analyser (Thermo Finnigan; Thermo Scientific, Hemel Hempstead, UK) connected to an isotope ratio mass spectrometer (Delta V Advantage; Thermo Scientific).

**Isolation of myofibrillar protein fractions and protein-bound alanine and $^{13}C_6$ phenylalanine enrichment.** Myofibrillar proteins were extracted by homogenizing 20–30 mg of muscle in ice-cold homogenization buffer [50 mM Tris·HCl (pH 7.4), 50 mM NaF, 10 mM $\beta$-glycerophosphate disodium salt, 1 mM EDTA, 1 mM EGTA and 1 mM activated $Na_3VO_4$ and a complete protease inhibitor cocktail tablet (Roche, West Sussex, UK) at 10 $\mu$L $\mu g^{-1}$ tissue and shaken for 10 min. Homogenates were spun at 1000 $g$ for 5min at 4°C and the supernatant was collected. The myofibrillar fraction of the pellet was solubilized for 30 min at 37°C in 0.3 M NaOH and separated from the insoluble collagen fraction by centrifugation. Myofibrillar proteins were precipitated using 1 M PCA and spun down at 3200 × $g$ for 20 min at 4°C before being hydrolysed overnight in 1 mL of 0.1 M HCl and 1 mL of Dowex $H^+$ Resin. Proteins were eluted from the resin using 2 M $NH_4OH$ and dried down at 70°C under a constant nitrogen flow. Amino acids for $^{13}C_6$ Phenylalanine enrichment were derivatized as their n-acetyl-n-propylester, and labelling was determined using a Thermo Delta V isotope ratio mass spectrometer with a Thermo GC ultra and PAL auto-sampler, Thermo GC Combustion III interface and Conflow IV interface. Amino acids for incorporation of deuterium into muscle-bound alanine were derivatized as their N-methoxycarbonyl methyl esters (Wilkinson et al., 2014). Labelling was determined using chromatography: pyrolysis: isotope ratio MS (Delta V Advantage) and ran alongside a standard curve of known DL-alanine-2.3.3.3-d4 enrichment to ensure measurement accuracy of the machine.

### Total and phosphorylated protein analysis

Western blot analyses were performed on the sarcoplasmic fraction obtained during myofibrillar protein extraction, as previously described. (Smeuninx et al., 2021) The following primary antibodies were used; total mechanistic target of rapamycin (mTOR; CST2983, 1:1000 in 5% BSA TBST), phospho-mTOR[Ser2448] (CST2971, 1:1000 in 5% BSA TBST), total protein kinase B (Akt; CST9272, 1:1000 in 5% BSA TBST), phospho-Akt[Ser473] (CST4060, 1:1000 in 5% BSA TBST), total eukaryotic elongation factor 2 (eEF2; CST2332, 1:1000 in 5% BSA TBST) and

phospho eEF2$^{Thr56}$ (CST2331, 1:1000 in 5% BSA TBST) purchased from Cell Signaling Technology (Hitchin, UK). The following HRP-linked anti-rabbit (CST7074) IgG dilutions were used: 1:5000 in 3% BSA TBST dilution for total and phospho mTOR$^{Ser2448}$, and 1:10,000 in TBST dilution for total Akt, phospho-Akt$^{Ser473}$, total eEF2 and phospho-eEF2$^{Thr56}$. Imaging was undertaken using a G:Box Chemi-XR5 (Syngene, Cambridge, UK), and bands were quantified using ImageJ (US National Institutes of Health, Bethesda, MD, UK). Relative arbitrary units were normalized to the total amount of protein loaded as visualized via Ponceau S staining. Specifically, bands were normalized to the ∼45 kDa actin band of Ponceau S stain as a loading control. No difference was detected between the ∼45 kDa actin band of Ponceau S stain across all samples. After these corrections, the phosphorylation of proteins as a proxy of their activation was expressed relative to the total abundance of protein.

### Gene expression analysis

Gene expression analysis was performed as previously described (Shad et al., 2019). Briefly, RNA was isolated from ∼20 mg of frozen powdered muscle homogenized in 1 mL of TRI Reagent (Sigma Aldrich, Gillingham, UK) and 200 $\mu$L of chloroform added to achieve phase separation. The RNA-containing supernatant was removed and purified using Reliaprep spin columns (Promega, Madison, WI, USA). RNA concentration and purity (ratio of the absorbance at 260 and 280 nm and was ≥1.85 for all samples) was determined using a FLUOstar Omega microplate reader. Then, 700 ng of total RNA was reverse-transcribed to cDNA in 20 $\mu$L volumes using the nanoScript 2 RT kit in combination with oligo(dT) and random primers (Primerdesign, Southampton, UK). cDNA was diluted to 5 ng $\mu$L$^{-1}$ prior to RT-qPCR analysis. All analyses were performed in triplicate using Primerdesign custom-made primer sequences, or commercially available 18S, B2M, GAPDH and ACTB. cDNA was added at5 and 20 ng, respectively, for housekeeping and human genes of interest to a 20 $\mu$L reaction volume. Thermal cycling conditions consisted of 2 min at 95°C, followed by 40 cycles of 10 s at 95°C and 60 s at 60°C. A melt curve was performed (Applied Biosystems, Thermo Fisher, UK) after qPCR to assure primer specificity. Results were analysed using Thermo Fisher Connect (Thermo Fisher) and expressed as fold change relative to baseline using the $2^{-\Delta\Delta CT}$ method. Data were normalized to the geometric mean of GAPDH, 18S and ACTB to minimize variation of the individual housekeeping genes.

All gene expression results are presented for $n = 9$ in each group as insufficient tissue was available for one participant. Gene expression targets were measured in muscle biopsy tissue obtained in the postabsorptive state.

### Fibre-type cross-sectional area

Muscle cross-sections (5 $\mu$m) were permeabilized in 0.02% Triton X-100 for 5 min before being incubated for 90 min in 5% normal goat serum. Subsequently, muscle cross-sections were incubated overnight in myosin heavy chain type I (IgG2b, BAF8, DSHB, Iowa City, IA, USA) and myosin heavy chain type II (IgG1, SC.71, DSHB) primary antibodies. On the following day, muscle cross-sections were washed three times for 5 min in 1× phosphate-buffered saline (PBS) and incubated in their respective secondary antibodies with WGA Igg for 90 min. Finally muscle cross-sections were washed in 1× PBS and slides mounted with Prolong Gold anti-fade reagent (P36930, Invitrogen). Images were captured using with an Eclipse E600 (Nikon, Badhoevedorp, the Netherlands) and a 20× zoom. All images were analysed using ImageJ Fiji software.

### Plasma hormone and analyte concentrations

Plasma insulin concentrations were analysed using a commercially available enzyme-linked immunosorbent assay according to the manufacturer's instructions (R&D Systems, Minneapolis, MN, USA). Plasma glucose was measured using a Roche Cobas 8000 analyser (Roche Diagnostics, Basel, Switzerland). Serum total cholesterol, High-density lipoprotein cholesterol (HDL-C), Triglyceride (TG) and Non esterified fatty acids (NEFA) (Randox, London, UK, for all) were analysed using an ILAB 650 Clinical Chemistry Analyser. Plasma creatine kinase and myoglobin were analysed using a Cobas 6000 E-module (Roche Diagnostics GmbH, Mannheim, Germany).

### Calculations

The iMyoPS and aMyoPS fractional synthetic rate (FSR) was determined as percentage per hour and percentage per day for [$^{13}$C$_6$] phenylalanine and [$^2$H] alanine, respectively, with the use of the precursor-product equation, as previously described (Smeuninx et al., 2017; Wilkinson et al., 2014). The use of tracer-naïve participants allowed us to use the pre-infusion mixed plasma protein $^{13}$C$_6$ phenylalanine enrichment as a proxy for basal muscle protein enrichment for measurement of postabsorptive aMyoPS rates. This approach has been validated in older individuals (Burd et al., 2012).

### Statistical analyses

Sample size was determined *a priori* based on effect size estimates from studies investigating the change in leg muscle mass with short-term disuse in older individuals (Dirks et al., 2014; Reidy et al., 2017; Tanner et al., 2015). Calculations were based on the

assumption that the loss in MRI-based quadriceps CSA would be largely attenuated in EX compared with CTL, as per comparator groups used in selected studies (Reidy et al., 2017; Tanner et al., 2015). Therefore, we determined that for a power of 80%, a sample size of $n = 10$ would allow us to detect differences of changes in quadriceps CSA between CTL and EX (G*Power [version 3.1.9.6], Heinrich Heine University Düsseldorf, Düsseldorf, Germany). Anthropometric, physical activity, dietary characteristics and blood hormone/analytes were analysed using a paired samples $t$ test. Quadriceps CSA, fibre type morphology, gene expression, and iMyoPS and aMyoPS were analysed using a two-way repeated-measures ANOVA (condition × time) with condition (CTL *vs.* EX) and time (pre-bed-rest *vs.* post-bed-rest). Body water $^2$H enrichment, plasma amino acid and $^{13}C_6$ phenylalanine were analysed using a one-way repeated-measures ANOVA with time as the within-subject factor. Delta and percentage change in quadriceps CSA, iMyoPS and aMyoPS for CTL and EX were analysed using a Student's $t$ test. Bonferroni *post hoc* tests were performed to correct for multiple comparisons when a significant main effect or interaction was identified. Correlations were assessed using Pearson's product moment correlation coefficient. All analyses were performed using Prism v. 5 (GraphPad Software, La Jolla, CA, USA). Significance was set at $P \leq 0.05$. All data are presented as mean ± SD.

## Results

### Anthropometric, physical activity and dietary characteristics

Participant body mass (~1.6%; $P = 0.022$), BMI (~1.7%; $P = 0.025$) and relative fat-free mass (~2.4%; $P = 0.044$) decreased from pre- to post-bed-rest, whereas relative fat mass increased (~6.7%; $P = 0.045$). Average daily step-count, and time spent performing light and moderate intensity activities were significantly lower, and sedentary time significantly greater, during bed-rest compared with habitual levels ($P < 0.001$ for all). Percentage daily vigorous activity was very low during the habitual phase and was not significantly altered during bed-rest. During bed-rest, dietary protein ($P = 0.020$), fibre ($P = 0.006$) and alcohol ($P < 0.001$) intake decreased significantly, whilst carbohydrate ($P = 0.007$) intake increased significantly compared with habitual levels, with no difference in total energy intake. Anthropometric, physical activity and dietary characteristics are presented in Table 1.

### Maximal strength and resistance training parameters

Estimated 1RM strength was 60.1 ± 11.0 and 54.8 ± 11.7 kg for the leg extension and leg curl machines, respectively. The average total RE volume was 5604 ± 1079 kg, performed at an average Borg CR-10 rating of 8.3 ± 0.7. Resistance exercise training prehabilitation data are presented in Table 2.

**Table 1. Participant anthropometric, activity and dietary characteristics before and during bed-rest**

| Characteristic | Habitual | Bed-rest | P-value |
|---|---|---|---|
| Age (years) | 72.0 ± 4.6 | | |
| Height (m) | 1.75 ± 0.07 | | |
| Weight (kg) | 82.2 ± 10.7 | 80.5 ± 11.1* | 0.022 |
| BMI (kg m$^{-2}$) | 26.8 ± 2.9 | 25.3 ± 2.7* | 0.025 |
| Body fat (%) | 25.8 ± 5.6 | 27.5 ± 5.4* | 0.045 |
| Fat-free mass (%) | 74.2 ± 5.6 | 72.5 ± 5.4* | 0.044 |
| Daily step-count | 8987 ± 2148 | 120 ± 195* | <0.001 |
| Sedentary activity (%) | 78.2 ± 5.58 | 94.4 ± 3.3* | <0.001 |
| Light activity (%) | 9.5 ± 1.8 | 3.1 ± 1.8* | <0.001 |
| Moderate activity (%) | 12.1 ± 5.0 | 2.5 ± 1.6* | <0.001 |
| Vigorous activity (%) | 0.15 ± 0.16 | 0.03 ± 0.06 | 0.103 |
| Total energy intake (kcal) | 2124 ± 392 | 2216 ± 389 | 0.589 |
| Protein intake (g kg$^{-1}$ day$^{-1}$) | 1.20 ± 0.28 | 0.95 ± 0.18* | 0.020 |
| CHO intake (g kg$^{-1}$ day$^{-1}$) | 2.05 ± 0.79 | 3.02 ± 0.63* | 0.007 |
| Fat intake (g kg$^{-1}$ day$^{-1}$) | 1.02 ± 0.30 | 1.21 ± 0.20 | 0.089 |
| Fibre intake (g kg$^{-1}$ day$^{-1}$) | 0.27 ± 0.12 | 0.10 ± 0.05* | 0.006 |
| Alcohol intake (g kg$^{-1}$ day$^{-1}$) | 0.19 ± 0.17 | 0.00 ± 0.00* | <0.001 |

Step-count, physical activity and dietary intake data are daily averages obtained over 5 days of bed-rest. BMI: body mass index; ALM: appendicular lean mass; SPPB: Short Physical Performance Battery. *Significant difference from corresponding pre-bed-rest value ($P < 0.05$). Values are means ± SD for $n$ = 10 participants.

**Table 2. Leg strength and resistance exercise parameters for leg extension and leg curl exercises**

| Parameter | Leg extension | Leg curl |
|---|---|---|
| Estimated 1RM (kg) | 60.1 ± 11.0 | 54.8 ± 11.7 |
| Average load per set (kg) | 40.2 ± 6.4 | 36.9 ± 8.4 |
| Total load (kg) | 240.7 ± 75 | 221.1 ± 50.6 |
| Average repetitions per set | 12.1 ± 0.7 | 12.2 ± 0.9 |
| Total repetitions | 72.4 ± 4.3 | 73.4 ± 5.2 |
| Average volume per set (kg) | 494 ± 106 | 452 ± 115 |
| Total volume (kg) | 2892 ± 447 | 2711 ± 689 |
| T-U-T per set (s) | 23.8 ± 4.9 | 23.5 ± 7.3 |
| T-U-T total (s) | 143.3 ± 29.5 | 140.7 ± 43.5 |
| Average Borg CR-10 | 8.7 ± 0.9 | 7.9 ± 1.1 |

1RM: one-repetition maximum strength, T-U-T: time under tension. Values are means ± SD for $n$ = 10 participants.

**Table 3. Quadriceps and fibre CSA for CTL and EX legs measured before and following 5 days of bed-rest**

| | CTL | | | EX | | |
|---|---|---|---|---|---|---|
| | Pre-bed-rest | Post-bed-rest | *P* value | Pre-bed-rest | Post-bed-rest | *P* value |
| Quadriceps CSA 20% (mm²) | 4630 ± 594 | 4612 ± 622 | 0.134 | 4682 ± 630 | 4670 ± 545 | 0.095 |
| Quadriceps CSA 40% (mm²) | 6607 ± 651 | 6533 ± 514** | 0.006 | 6804 ± 667 | 6775 ± 530*# | 0.026 |
| Quadriceps CSA 60% (mm²) | 7008 ± 749 | 6778 ± 658** | 0.002 | 7183 ± 704 | 7050 ± 646**## | 0.007 |
| Quadriceps CSA 80% (mm²) | 4993 ± 702 | 4869 ± 586*** | <0.001 | 5116 ± 625 | 5023 ± 621** | 0.002 |
| Type I fibre CSA (μm²) | 6159 ± 1409 | 6050 ± 1002 | 0.322 | 5610 ± 1169 | 5751 ± 2061 | 0.629 |
| Type II fibre CSA (μm²) | 5544 ± 1607 | 5465 ± 2275 | 0.606 | 4665 ± 1313 | 4854 ± 1419 | 0.522 |

Values are means ± SD for *n* = 10 participants for quadriceps CSA and *n* = 8 participants for fibre CSA. Asterisks indicate significantly different from corresponding pre-bed-rest value (*$P < 0.05$, **$P < 0.01$, ***$P < 0.001$), and hash symbols indicate significantly different from CTL (#$P < 0.01$, ##$P < 0.001$).

## Quadriceps cross-sectional area and fibre-type morphology

Pre-bed-rest (baseline) quadriceps CSA did not differ between CTL and EX at 20%, 40%, 60% or 80% of the distance measured between the top of the patella and greater trochanter. Quadriceps CSA decreased significantly from pre- to post-bed-rest at 40%, 60% and 80% of lower limb length in CTL ($P = 0.006$, 0.002 and <0.001, respectively) and EX ($P = 0.026$, 0.007 and 0.002, respectively). No change in quadriceps CSA was observed in CTL or EX at 20% of lower limb length ($P = 0.134$ and 0.095, respectively). The relative decrease in quadriceps CSA from pre- to post-bed-rest at 40% and 60% of lower limb length was significantly attenuated in EX (~0.43% and 1.85%, respectively) compared with CTL (~1.12% and 3.28%, respectively; $P < 0.001$ and $P = 0.0125$ for 40% and 60%, respectively). No significant leg, time or interaction effects were found for fibre-type CSA. The magnitude of change in fibre-type CSA from pre- to post-bed-rest did not differ between CTL and EX. Quadriceps and fibre-type CSA data are presented in Table 3 and Fig. 2.

## Blood hormones and analytes

Fasted plasma insulin and serum triglyceride concentrations increased significantly from pre- to-post-bed-rest ($P < 0.01$ for both). Fasted serum total cholesterol and serum HDL-C concentration decreased significantly from pre- to post-bed-rest ($P < 0.01$ for both). Fasted serum non-HDL, serum NEFA, total cholesterol:HDL-C ratio, plasma creatine kinase and plasma myoglobin did not differ from pre- to post-bed-rest. Blood hormone and analyte data are presented in Table 4.

## Plasma amino acid, ¹³C₆ phenylalanine and ²H body water enrichment

Plasma leucine concentrations were significantly elevated above postabsorptive values 20, 40, 60, 90, 120 and 180 min after drink consumption (Fig. 3*A*), whilst plasma phenylalanine concentrations were significantly elevated

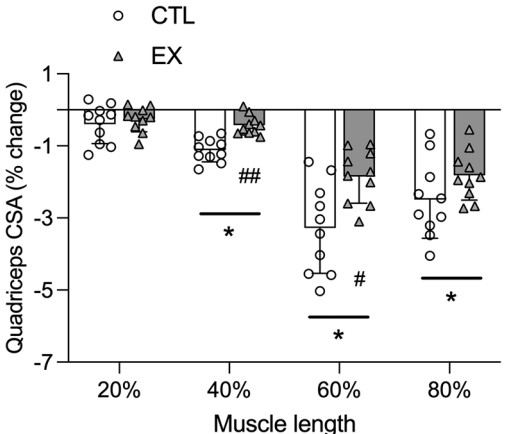

**Figure 2. Percentage change in quadriceps cross-sectional area during 5 days of bed-rest in healthy older males in a leg that had undergone a bout of resistance exercise the evening prior to bed-rest (EX) or the contralateral non-exercised control leg (CTL)**

Magnetic resonance imaging was obtained at 20%, 40%, 60% and 80% of the length between the top of the patella and the greater trochanter (distal to proximal). Between-leg differences were analysed using a Student's paired *t* test. Values are means ± SD (*n* = 10 for all panels). Significance was set at $P < 0.05$. A significant decrease in quadriceps CSA was noted at 40%, 60% and 80% of muscle length (*$P < 0.01$). The decrease in quadriceps CSA at 40% and 60% of muscle length was significantly attenuated in EX compared with CTL (#$P < 0.05$, ##$P < 0.001$).

**Table 4. Fasting blood hormone and analyte concentrations before and following 5 days of bed-rest**

| | Pre-bed-rest | Post-bed-rest | *P*-value |
|---|---|---|---|
| Plasma insulin (pmol L$^{-1}$) | 32.8 ± 8.0 | 39.0 ± 13.3* | 0.009 |
| Serum total cholesterol (mmol L$^{-1}$) | 4.87 ± 1.04 | 4.50 ± 1.00* | 0.006 |
| Serum HDL-C (mmol L$^{-1}$) | 1.51 ± 0.35 | 1.23 ± 0.26* | 0.003 |
| Serum non-HDL-C (mmol L$^{-1}$) | 3.36 ± 0.85 | 3.15 ± 0.90 | 0.061 |
| Total cholesterol:HDL-C ratio | 3.29 ± 0.56 | 3.47 ± 0.69 | 0.163 |
| Serum NEFA (mmol L$^{-1}$) | 0.51 ± 0.23 | 0.46 ± 0.23 | 0.511 |
| Serum triglycerides (mmol L$^{-1}$) | 1.05 ± 0.38 | 1.40 ± 0.49* | 0.008 |
| Plasma creatine kinase (IU L$^{-1}$) | 172.6 ± 41.7 | 160.1 ± 43.8 | 0.507 |
| Plasma myoglobin (ng·ml$^{-1}$) | 52.2 ± 9.5 | 50.9 ± 11.0 | 0.733 |

HDL-C: high-density lipoprotein cholesterol; NEFA: non-esterified fatty acids. Values are means ± SD for *n* = 9 participants. *Significantly different from corresponding pre-bed-rest value (*P* < 0.05).

above postabsorptive values 20, 40, 60 and 90 min after drink consumption (Fig. 3*B*). Body water $^2$H enrichment, assessed via saliva, was 0.31 ± 0.01 APE at 24 h after the first bolus D$_2$O dose and averaged 0.28 ± 0.004 APE over the entire study duration (Fig. 3*C*). Linear regression analysis indicated that the slope of the body water enrichment curve was significantly different from zero ($r^2 = 0.90$; $P < 0.001$).

The slight decline in body $^2$H water enrichment is accounted for in the precursor-product calculation

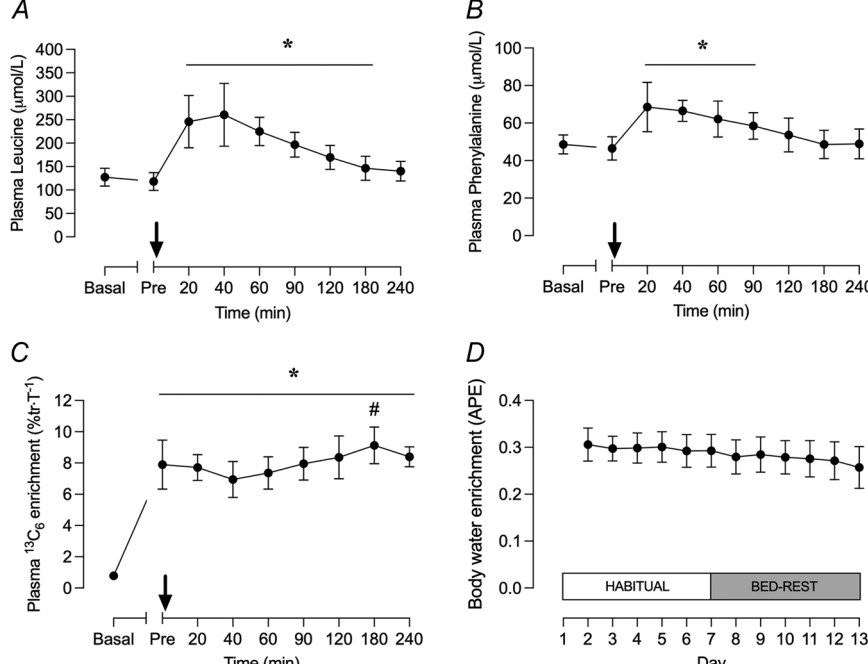

**Figure 3. Amino acid concentrations and stable isotope enrichments in plasma and saliva**
Plasma leucine concentration (*A*), phenylalanine concentration (*B*) and $^{13}$C$_6$ phenylalanine enrichment (*C*) measured in the experimental infusion trial (Day 13) and body water $^2$H enrichment (*D*) measured in daily saliva samples over the course of the study (Days 1–13). In *A–C*, the black arrow indicates the point at which a 15 g milk protein isolate bolus was consumed. Plasma and saliva data were analysed using a one-way repeated-measures ANOVA with time as the within-subject factor. Values are means ± SD (*n* = 10 for all panels). Significance was set at *P* < 0.05. Plasma leucine and phenylalanine concentrations were significantly increased above basal-fasted and pre-drink values at 20–120 min and 20–90 min after drink, respectively (*P* < 0.05 for both). Plasma $^{13}$C$_6$ phenylalanine enrichment was significantly increased above basal-fasted values after drink and remained elevated for the duration of the experiment (*P* < 0.05). Plasma $^{13}$C$_6$ phenylalanine enrichment at 180 min after drink was significantly greater than at 20, 40, 60 and 90 min ($^\#P$ < 0.05).

and had no bearing on iMyoPS rates calculated over habitual and bed-rest phases fort EX and CTL. Plasma $^{13}C_6$ phenylalanine enrichment increased significantly above basal values 180 min after initiation of the stable isotope tracer infusion and remained elevated for the duration of the trial ($P < 0.001$; Fig. 3D). Plasma $^{13}C_6$ phenylalanine enrichment 180 min after drink consumption was significantly different to values at 20, 40, 60 and 90 min. Linear regression analysis revealed that the $^{13}C_6$ phenylalanine enrichment slopes in both groups were not significantly different from zero.

### Integrated myofibrillar protein synthesis (iMyoPS)

There was no significant time × leg interaction effect ($P = 0.155$). A significant main effect for time ($P = 0.031$) and leg ($P = 0.020$) was observed (Fig. 4A). Habitual rates of iMyoPS did not differ between CTL and EX ($1.28 \pm 0.038$ and $1.36 \pm 0.044\%$ day$^{-1}$, respectively; $P = 0.595$). Rates of iMyoPS decreased significantly during bed-rest compared with habitual values in CTL ($1.08 \pm 0.04\%$ day$^{-1}$; $P = 0.022$), but not EX ($1.27 \pm 0.065\%$ day$^{-1}$; $P = 0.487$). Rates of iMyoPS over bed-rest were significantly lower in CTL compared with EX ($P = 0.029$). The change in iMyoPS over bed-rest from habitual values was not significantly different between EX and CTL ($P = 0.224$; Fig. 4B). The change in iMyoPS over bed-rest was correlated with the peak change in quadriceps CSA at 60% of muscle length in

CTL ($P = 0.009$, $r^2 = 0.529$) but not EX ($P = 0.584$, $r^2 = 0.039$, Fig. 4C).

### Acute myofibrillar protein synthesis (aMyoPS)

There was no significant time × leg interaction effect ($P = 0.174$) nor any significant main effect of leg ($P = 0.189$). A significant main effect for time ($P = 0.001$) was observed (Fig. 5A). Postabsorptive rates of aMyoPS did not differ between CTL and EX ($0.030 \pm 0.003$ and $0.035 \pm 0.004\%$ h$^{-1}$, respectively; $P = 0.155$). Postprandial aMyoPS rates were significantly increased above postabsorptive values in EX ($0.041 \pm 0.005\%$ h$^{-1}$, $P = 0.033$), but not CTL ($0.033 \pm 0.003\%$ h$^{-1}$; $P = 0.579$). The postprandial change in aMyoPS from the postabsorptive state did not differ between EX and CTL (Fig. 5B). The absolute difference in integrated and acute MyoPS between CTL and EX was similar [iMyoPS: $-0.12 \pm 0.08\%$ day$^{-1}$, aMyoPS: $-0.15 \pm 0.11\%$ day$^{-1}$ (assuming two-thirds of the day is spent fasted and one-third fed), $d = 0.008$, Fig. 5C].

### Gene and protein expression

There was no significant time × leg interaction effect for myostatin expression ($P = 0.370$). There was a significant main effect of time ($P = 0.013$) or leg ($P = 0.044$) for myostatin expression (Fig. 6A). Myostatin expression increased from pre- to post-bed-rest in CTL

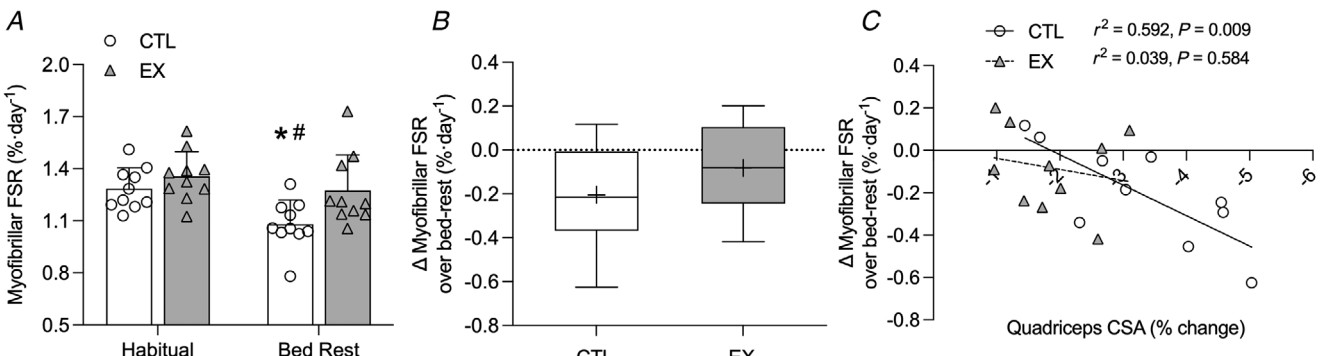

**Figure 4. Integrated rates of myofibrillar protein synthesis**
*A*, integrated myofibrillar protein fractional synthesis rates (FSR) over the course of 7 days before (habitual) and during 5 days of bed-rest in exercised (EX) and non-exercised (CTL) legs in healthy older males. *B*, delta change in integrated myofibrillar protein fractional synthesis rates from prehabilitation to bed-rest in EX and CTL. C, correlation between the change in iMyoPS and change in quadriceps cross-sectional area at 60% of muscle length. Values in *A* are means ± SD and individual participant data (*n* = 10). In *B*, the boxes represent the 25th to 75th percentiles, error bars represent minimum and maximum values, and horizontal lines and crosses within boxes represent median and mean values, respectively (*n* = 10). Values in *C* are individual participant data (*n* = 10). Data in *A* were analysed using a two-way repeated-measures ANOVA (condition × time) with condition (CTL *vs.* EX) and time (habitual *vs.* bed-rest). Data in *B* were analysed using a Student's *t* test. Data in *C* were analysed by Pearson's *product moment* correlation coefficient. Bonferroni *post hoc* tests were performed to correct for multiple comparisons when a significant main or interaction effect was identified. Significance was set at *P* < 0.05. Rates of iMyoPS declined significantly over bed-rest compared with habitual values in CTL only (*$P$ < 0.05). Rates of iMyoPS over bed-rest were significantly lower in CTL compared with EX (#$P$ < 0.05). The change in iMyoPS over bed-rest was inversely correlated with the change in quadriceps CSA at 60% of muscle length in CTL, but not EX.

only ($P = 0.011$). There was no significant time × leg interaction effect for MAFbx expression ($P = 0.847$). There was a significant effect of time ($P = 0.004$) and leg ($P = 0.003$) for MAFbx expression (Fig. 6B). MAFbx expression increased from pre- to post-bed-rest in CTL ($P = 0.028$) and EX ($P = 0.042$), with no difference between legs. There was no significant time × leg interaction effect for MuRF-1 expression ($P = 0.519$), nor any significant main effect of time ($P = 0.317$). There was a main effect of leg ($P = 0.033$) for MuRF-1 expression (Fig. 6C). No significant interaction effect or main effect of time and leg were found for p70S6K or mTOR expression (Fig. 6D and E). No significant group, time or interaction effects were found for Akt$^{ser473}$, mTOR$^{ser2448}$ or eEF2$^{Thr56}$ protein expression between habitual and post-bed-rest phases or between postabsorptive to postprandial states (Fig. 7A–F).

## Discussion

Concise RE intervention prior to disuse events in older individuals offers a cost-effective way to attenuate muscle mass loss by capitalizing on better patient health status, with implications for sarcopenia progression and healthcare expenditure in our ageing populations (Baker, 2020; Sousa et al., 2016). The present study demonstrates that a single bout of unilateral leg resistance exercise (EX) performed the evening prior to 5 days of bed-rest attenuated the decline in iMyoPS and quadriceps CSA

that was observed in the non-exercised leg (CTL) of older individuals. The change in iMyoPS was correlated with the peak loss of quadriceps CSA in CTL only. The attenuated rate of decline in iMyoPS and muscle atrophy in EX may be explained by preserved postprandial aMyoPS stimulation.

The removal of muscle contractile activity in older individuals undergoing bed-rest results in blunted postprandial (Tanner et al., 2015) and postabsorptive aMyoPS rates (Kortebein et al., 2007), which underlines the reduction in iMyoPS rates reported during disuse events (McGlory et al., 2018; Shad et al., 2019). RE can robustly increase aMyoPS in both the postabsorptive (Biolo et al., 1995; Kumar et al., 2012) and the postprandial state through increasing the utilization of dietary-derived amino acids (Pennings et al., 2011; Yang et al., 2012). RE-induced increases in aMyoPS are reportedly sustained for 24–72 h after exercise (Burd et al., 2011; McKendry et al., 2019; Miller et al., 2005). We and others have previously shown that a single bout of unilateral RE, similar to that used herein, increased iMyoPS rates over 48 h after exercise in older adults (McKendry et al., 2019). Given evidence that impairments in muscle proteostasis (particularly MyoPS) and the rate of muscle atrophy are most pronounced in the initial days of bed-rest (Di Girolamo et al., 2021; Hardy et al., 2022; Marusic et al., 2021; Tanner et al., 2015), we posited that the sustained muscle anabolic response to a prior bout of unilateral RE in EX would offset the decline in muscle mass. As hypothesized, rates of iMyoPS over bed-rest

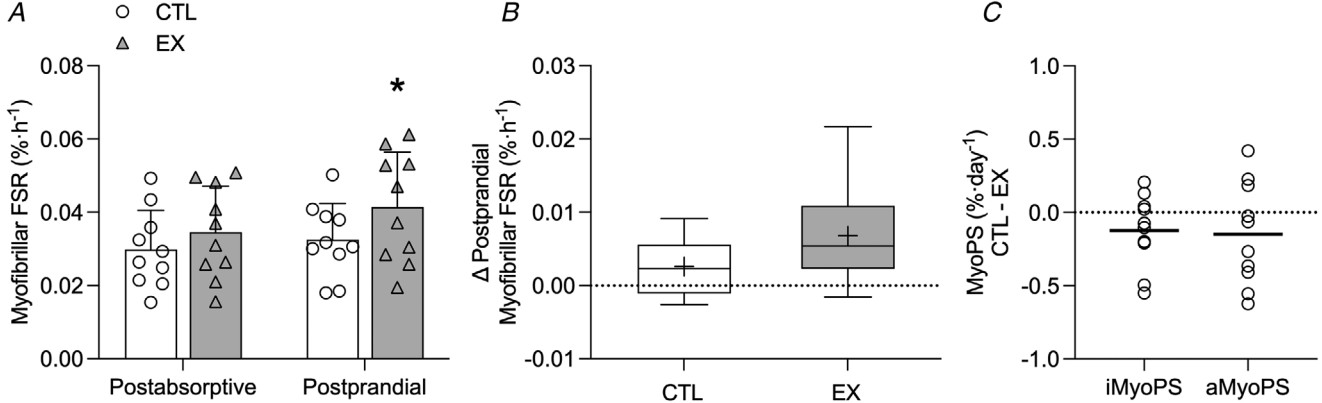

**Figure 5. Acute rates of myofibrillar protein synthesis**
*A*, acutely measured myofibrillar protein synthesis rates (FSR) in the postabsorptive and postprandial state, after ingestion of 15 g of milk protein, in EX and CTL. *B*, delta change in acute myofibrillar protein fractional synthesis rates from postabsorptive to postprandial state in EX and CTL. *C*, the absolute change in integrated (iMyoPS) or acute myofibrillar protein synthesis (assuming two-thirds of the day is spent fasted and one-third fed). Values in *A* are means ± SD and individual participant data (*n* = 10). In *B*, the boxes represent the 25th to 75th percentiles, error bars represent minimum and maximum values, and horizontal lines and crosses within boxes represent median and mean values, respectively (*n* = 10). In *C*, circles represent individual data points, and the horizontal line represents the mean (*n* = 10). Data in *A* were analysed using a two-way repeated-measures ANOVA (condition × time) with condition (CTL *vs*. EX) and time (habitual *vs*. bed-rest). Data in *B* and *C* were analysed using a Student's *t* test. Bonferroni *post hoc* tests were performed to correct for multiple comparisons when a significant main or interaction effect was identified. Significance was set at $P < 0.05$. Rates of aMyoPS were significantly increased from the postabsorptive to postprandial state in EX (*$P < 0.05$).

were not different from habitual values in EX (−6%) but significantly declined in CTL (−16%). Although we may have been underpowered to detect a significant interaction effect, iMyoPS rates over bed-rest were significantly lower (−15%) in CTL compared with EX.

The divergence in iMyoPS rates between CTL and EX during bed rest probably explains the attenuated quadriceps muscle loss at 40% and 60% of thigh length in EX compared with CTL. This observation is supported by a recent study in younger adults, showing that a single bout of prior damaging exercise (300 eccentric contractions) prevented the decline in iMyoPS during 7 days of limb immobilization, and the decline in quadriceps volume in the first 2 days of immobilization. These observations remained when the authors corrected for

damaged-induced oedematous swelling of both legs and changes in volume to the non-immobilized control leg. Although we did not measure muscle damage directly, the current load–volume of our RE protocol was likely to have elicited only low to mild damage symptoms in our older cohort (Fernandes et al., 2021). Indeed, in a subset of eight participants we used B-mode ultrasound to measure thickness of the vastus lateralis and summed quadriceps muscles at 50% of limb length as a proxy for oedema (Dourado et al., 2023). We did not detect any increase in quadriceps muscle thickness at 18 h after RE (in close proximity to the baseline MRI scan) or following bed-rest that would suggest oedema was present and could have influenced the attenuation of muscle atrophy in EX compared with CTL (data not shown).

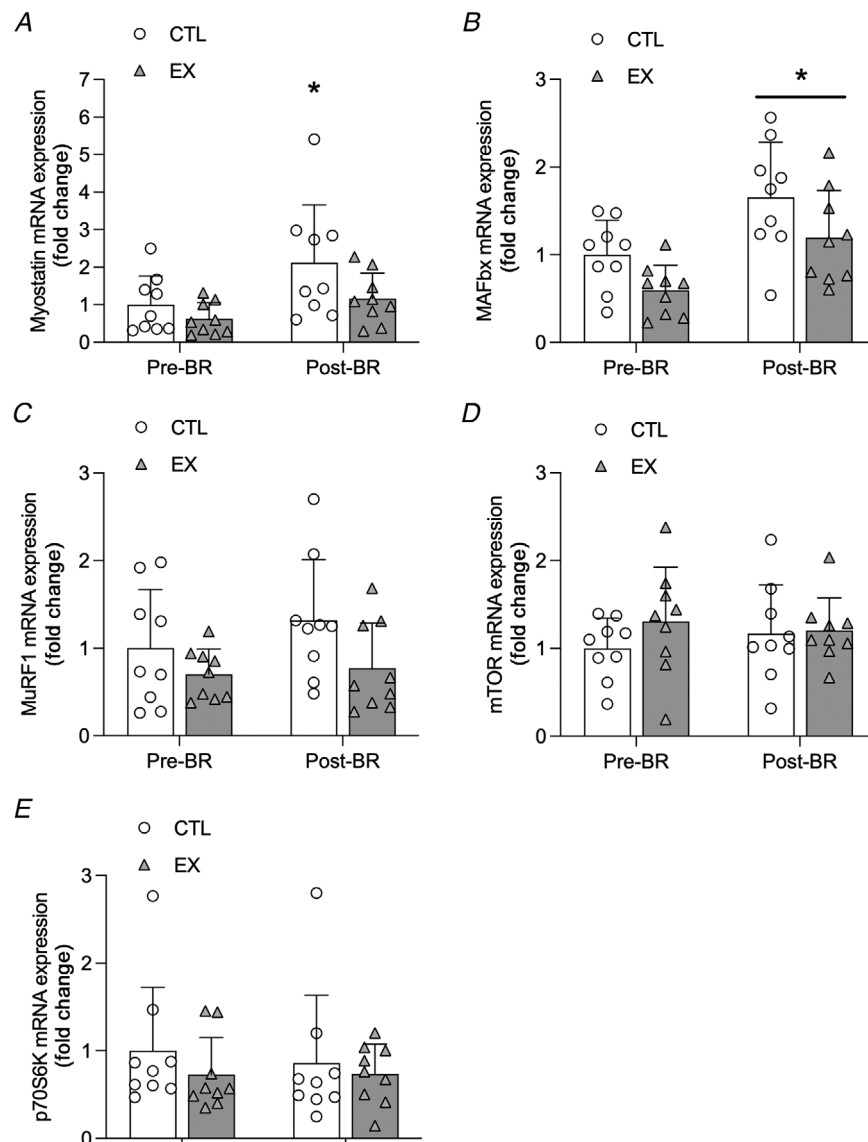

**Figure 6. Gene expression of anabolic and catabolic signaling**

*A–E*, mRNA expression of myostatin (*A*), MAFbx (*B*), MuRF1 (*C*), mTOR (*D*) and p70S6K (*E*) immediately before (Pre-BR) and following 5 days of bed-rest (Post-BR) in older men. Data are expressed as the fold-change from levels measured in CTL at Pre-BR, which was normalized to a value of 1. All targets were measured in muscle biopsy tissue obtained in the postabsorptive state. Values are means ± SD and individual participant data (*n* = 9 for all targets). Significance was set at *P* < 0.05. There was a significant increase in myostatin mRNA expression post-BR compared with pre-BR for CTL only (*P* < 0.05). MAFbx mRNA expression was significantly increased post-BR compared with pre-BR for EX and CTL (*P* < 0.05) with no difference between legs.

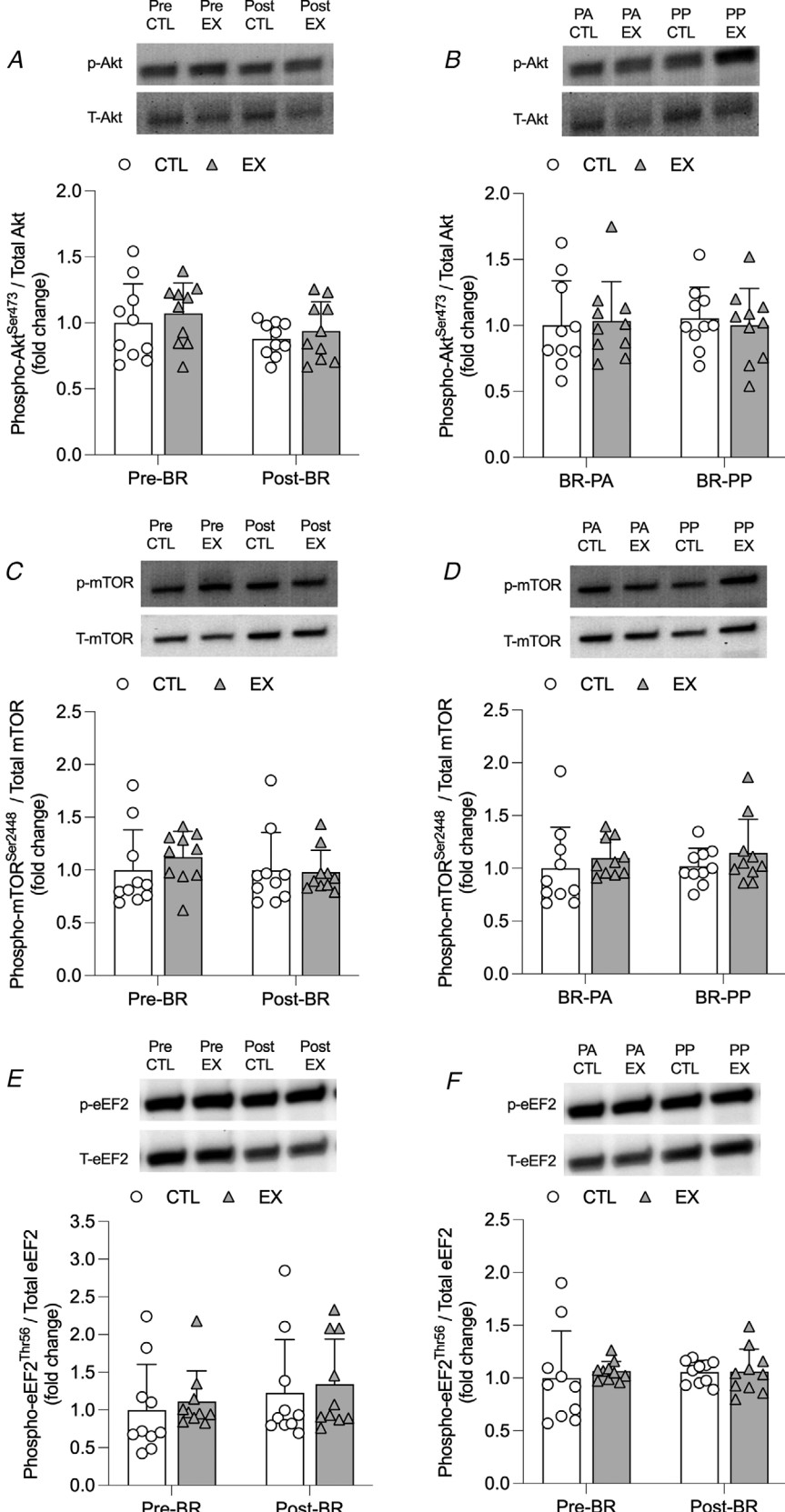

**Figure 7. Protein expression of anabolic signaling**

*A–F*, changes in protein expression of Akt$^{S473}$ (*A–B*), mTOR$^{S2448}$ (*C–D*) and eEF2$^{T56}$ (*E–F*) following 5 days of bed-rest in older men with (EX) or without (CTL) a prior bout of resistance exercise prehabilitation. Protein phosphorylation for all targets is presented relative to respective total protein expression. *A*, *C* and *E* show protein expression in EX and CTL in the habitual and post-bed-rest phases and are expressed as the fold-change from levels measured in control (CTL) in the habitual phase, which was normalized to a value of 1 (all targets were measured in a postabsorptive state). *B*, *D* and *F* show protein expression in EX and CTL after bed-rest in the postabsorptive (BR-PA) or 4 h postprandial state after ingestion of 15 g of milk protein (BR-PP) and are expressed as the fold-change from levels measured in CTL after bed-rest, which was normalized to a value of 1. Values in *A–F* are means ± SD and individual participant data (*n* = 9 for all targets). Significance was set at *P* < 0.05. No significant group, time or interaction effects were found for Akt$^{ser473}$, mTOR$^{ser2448}$ or eEF2$^{Thr56}$ protein expression between habitual and post-bed-rest phases or between BR-PA to BR-PP. Representative western blot lanes may appear duplicated across panels at post-BR and BR-PA time-points, as these images represent the same data point compared against habitual or BR-PP (in adjacent panels).

Furthermore, circulating markers of muscle damage, plasma creatine kinase and myoglobin did not differ from pre- to post-bed-rest. Despite the clear decline in quadriceps CSA across legs, and attenuation of disuse atrophy in EX, we were unable to detect any significant changes in fibre-type size with bed-rest or differences between EX and CTL. This finding is probably explained by the considerable inter- and intra-individual variation in fibre CSA (Horwath et al., 2021). Similarly, Wall et al. (2014) reported no change in fibre-type CSA with 5 days of knee immobilization, despite a significant decline in quadriceps CSA. Therefore, we can be confident that the present findings show that a feasible and accessible bout of RE attenuates the decline in iMyoPS and muscle mass in older adults during short-term bed-rest.

The decline in iMyoPS and muscle mass during bed-rest is underlined by diminished aMyoPS rates (Drummond et al., 2012; English et al., 2016). Prior RE could therefore provide a stimulus to bolster/retain aMyoPS rates during bed-rest, particularly the muscle anabolic response to feeding occasions that is known to decline rapidly with disuse (Kilroe et al., 2019; Tanner et al., 2015). The potential for RE to support postabsorptive and postprandial aMyoPS was investigated after the final day of bed-rest. No difference in postabsorptive rates of aMyoPS or regulatory protein signalling (mTOR, Akt and eEF2) were observed between EX and CTL. However, ingestion of 15 g of milk protein isolate ($\sim$0.21 g kg$^{-1}$ body weight) significantly increased aMyoPS above postabsorptive values in EX (17%) but not CTL (10%). Furthermore, the absolute difference in aMyoPS between CTL and EX was similar to the absolute difference in iMyoPS. The absence of a stimulatory effect of protein ingestion on aMyoPS and regulatory protein signalling in CTL is congruent with previous observations in older adults after 5 days of bed-rest (Tanner et al., 2015). Together, we speculate that a single bout of prior RE may have alleviated muscle anabolic resistance with 5 days of bed-rest in older individuals, but this requires further resolution along with underlying signalling mechanisms. We acknowledge that any potential effect of RE on postprandial aMyoPS stimulation may have dissipated over the course of bed-rest and could explain the absence of a significant interaction effect between EX and CTL legs at the time of assessment (6 days after RE). We were unable to measure aMyoPS rates prior to bed-rest and can only speculate on how aMyoPS may have been altered. However, given that quadriceps muscle mass loss was noted in both legs and regulatory signalling expression was not altered by protein ingestion, we suggest that bed-rest still probably impaired aMyoPS in EX.

The present study findings contrast with our earlier work showing that four bouts of RE (each identical to the present RE protocol) undertaken over 7 days failed to offset the decline in iMyoPS and muscle atrophy in

response to a subsequent 5 days of bed-rest in older adults (Smeuninx et al., 2021). Importantly, four bouts of prior RE increased iMyoPS rates over the 7 days prior to bed-rest in EX compared with CTL. It is also possible that the magnitude of the increase in aMyoPS and iMyoPS diminished with each subsequent RE bout (Damas et al., 2016; Wilkinson et al., 2008). Therefore, one might expect the withdrawal of contraction from EX during the subsequent bed-rest period to result in a relative decline in iMyoPS and muscle atrophy equivalent to that observed in CTL, which was the case. In contrast, as the single bout of RE in the present study was performed in the evening prior to bed-rest, iMyoPS rates at the onset of disuse were equivalent between legs. Hence, a single bout of prior RE provided a stimulus to augment/maintain iMyoPS in the initial days of bed-rest (potentially through preserved postprandial aMyoPS responsiveness) and was therefore more effective in attenuating muscle atrophy than four bouts of prior RE. Together, these findings suggest that the time-course and frequency of concise RE prehabilitation before disuse events is an important consideration for muscle-specific outcomes in older adults.

Whilst impaired aMyoPS is regarded as the principal mechanistic driver of disuse-induced muscle atrophy, a role for elevated muscle protein breakdown (MPB) has been implicated in the first several days of disuse (Tanner et al., 2015; Wall et al., 2014). In partial agreement with Tanner et al. (2015) we observed a significant increase in the mRNA expression of the ubiquitin ligase, MAFBX, after 5 days of bed-rest in EX and CTL, whereas MuRF-1 expression remained unchanged. The relevance of the increased MAFbx expression in CTL and EX is questionable given that quadriceps muscle atrophy with bed-rest was attenuated in EX compared with CTL. However, we did observe an increase in myostatin mRNA expression, a negative regulator of muscle mass associated with MPB (McFarlane et al., 2006), in CTL only, although this effect was not significantly different from EX. Based on these findings and the absence of confirmatory protein expression of the selected targets, it is not possible to define whether MPB differed between legs and influenced muscle loss over 5 days of bed-rest in older individuals. Given the differences in rates of iMyoPS and aMyoPS between EX and CTL, and the correlation between the change in iMyoPS and quadriceps CSA in CTL, it appears that deficits in muscle anabolic regulation underpin the observed quadriceps muscle loss with little to no appreciable role of elevated MPB. In support of this notion, we recently demonstrated that MPB was not altered over 4 days of uniteral limb immobilization in younger adults, whereas declines in MyoPS were associated with changes in muscle thickness (Brook et al., 2022).

In the present study, step-count during 5 days of bed-rest was $\sim$98% lower than habitual levels, and

sedentary time increased at the expense of a reduction in light- to moderate-intensity physical activity. The very low-level activity data reported herein are more stark than typical step-counts (Fisher et al., 2011; Sallis et al., 2015), sedentary/lying time and walking time (Brown et al., 2009) reported in older individuals during a typical in-patient stay, and more closely reflect older patients with functional limitations (Ostir et al., 2013). These dramatic alterations in physical activity during bed-rest resulted in a loss of quadriceps CSA at 40%, 60% and 80% of muscle length, comparable to previous observations (Smeuninx et al., 2021). However, a single bout of RE performed on the evening prior to bed-rest attenuated the decline in quadriceps muscle mass at 40% (0.43% *vs.* 1.12% respectively, for EX and CTL) and 60% of muscle length (1.85% *vs.* 3.28% for EX and CTL, respectively). It is possible that the RE protocol we used induced a neural and muscle anabolic (MyoPS) stimulus targeted to the mid-quadriceps, thereby explaining why muscle atrophy was attenuated in this region only. Indeed, others have reported regional differences in RE-induced neural activity and adaptive remodelling of the quadriceps (Balshaw et al., 2017; Ema et al., 2013; Narici et al., 1996). Manipulating RE parameters, such as contraction type, repetition duration, exercise selection and range of motion may provide a stimulus to attenuate muscle atrophy across all quadriceps sites (Fragala et al., 2019). It is worth noting that whilst iMyoPS rates were relatively well preserved over 5 days bed-rest in EX (e.g. not different from habitual values), muscle atrophy was not completely prevented, possibly due to the temporal change in iMyoPS over bed-rest. Specifically, prior RE may have elicited greater rates of iMyoPS during the early phase of bed-rest (i.e. Days 1–3), which declined significantly in the later phase (i.e. Days 3–5), such that the average daily iMyoPS rate over the entire 5 days of bed-rest was numerically, but not significantly, lower than habitual values. Nonetheless an important next step is to understand the functional and clinical implications of attenuated muscle atrophy with a single bout of RE in older patient populations. Due to the invasive procedures and logistically complexities of the study protocol, it was not possible to collect reliable strength/function data from participants following bed-rest.

In conclusion, the novel findings of the present study show that a single bout of prior RE offset the impairment in muscle anabolism and supports greater muscle mass retention in older individuals during a typical in-patient bed-rest period. Undoubtedly, longer-term multi-modal prehabilitation strategies are the most effective way to optimize patient state, buffer the decline in physiological function during disuse and improve clinical outcomes. Nonetheless, a single bout of prior RE could be particularly relevant when longer-term prehabilitation is not possible, for example with stand-by/short-notice surgery, lack of access or inability to exercise regularly.

Given the notion that RE may have overcome impairments in iMyoPS in the initial 24–72 h of bed-rest, where the rate of muscle atrophy is most pronounced, we suggest this strategy may not offer the same level of protection for iMyoPS and muscle mass with longer-term disuse events (>5 days). Future investigations should seek to understand the feasibility of implementing single-bout RE in older surgical patient populations and the impact on muscle mass, function and clinical outcomes.

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

## Additional information

### Data availability statement

The datasets generated during and/or analysed during the current study are not publicly available but are available from the corresponding author on reasonable request.

### Competing interests

None of the authors have any conflicts of interest to disclose.

### Author contributions

Conceptualization and design: B.S., A.B.R., A.P., P.J.A. and L.B. Funding acquisition: A.B.R., A.P., P.J.A. and L.B. Data acquisition: B.S., Y.E., E.S., P.T.M. and L.B. Formal analysis: B.S., P.T.M., M.K., A.E.B., A.P., M.S.B., N.G., D.W., K.S., P.J.A. and L.B. Original draft: B.S. and L.B. Review and editing: B.S., Y.E., E.S., A.B.R., P.T.M., M.K., A.E.B., A.P., M.S.B., N.G., D.W., K.S., P.J.A. and L.B. All authors have approved the final version of the manuscript and agree to be accountable for all aspects of the work. All persons designated as authors qualify for authorship, and all those who qualify for authorship are listed.

### Funding

This work was supported by an award from the Biotechnology and Biological Sciences Research Council to L.B. (BB/N018214/1) and the NIHR Clinical Research Facility (CRF) in University Hospitals Birmingham NHS Foundation Trust, Birmingham.

### Acknowledgements

The authors would like to thank the research participants for their time and effort in completing the study. The authors extend our appreciation to the CRF nursing staff for their support throughout the trial.

### Keywords

disuse, exercise training, muscle anabolism, sarcopenia

### Supporting information

Additional supporting information can be found online in the Supporting Information section at the end of the HTML view of the article. Supporting information files available:

**Peer Review History**

