## [Peer Review History · The Journal of Physiology]

A single bout of prior resistance exercise attenuates muscle atrophy and declines in myofibrillar protein synthesis during bed-rest in older men.

Benoit Smeuninx, Yasir S Elhassan, Elizabeth Sapey, Paul T Morgan, Marie Korzepa, Archie E Belfield, Alison Rushton, Andrew Philp, Nima Gharahdaghi, Matthew S Brook, Daniel James Wilkinson, Ken Smith, Philip J Atherton, and Leigh Breen

DOI: 10.1113/JP285130

Corresponding author(s): Leigh Breen (l.breen@bham.ac.uk)

Review Timeline:

Submission Date:	14-Jul-2023
Editorial Decision:	17-Aug-2023
Revision Received:	14-Sep-2023
Editorial Decision:	25-Sep-2023
Revision Received:	26-Sep-2023
Editorial Decision:	03-Oct-2023
Revision Received:	03-Oct-2023
Accepted:	05-Oct-2023

Senior Editor: Karyn Hamilton

Reviewing Editor: Christopher Sundberg

Transaction Report:

Dear Dr Breen,

Re: JP-RP-2023-285130 "A single bout of prior resistance exercise attenuates declines in myofibrillar protein synthesis and muscle atrophy during bed-rest in older men." by Benoit Smeuninx, Yasir S Elhassan, Elizabeth Sapey, Paul T Morgan, Alison Rushton, Andrew Philp, Nima Gharahdaghi, Matthew S Brook, Daniel James Wilkinson, Ken Smith, Philip J Atherton, and Leigh Breen

Thank you for submitting your manuscript to The Journal of Physiology. It has been assessed by a Reviewing Editor and by 2 expert referees and we are pleased to tell you that it is potentially acceptable for publication following satisfactory major revision.

LANGUAGE EDITING AND SUPPORT FOR PUBLICATION: If you would like help with English language editing, or other article preparation support, Wiley Editing Services offers expert help, including English Language Editing, as well as translation, manuscript formatting, and figure formatting at www.wileyauthors.com/eoo/preparation. You can also find resources for Preparing Your Article for general guidance about writing and preparing your manuscript at www.wileyauthors.com/eoo/prepresources.

REVISION CHECKLIST:

Please upload two versions of your manuscript text: one with all relevant changes highlighted and one clean version with no changes tracked. The manuscript file should include all tables and figure legends, but each figure/graph should be uploaded as separate, high-resolution files. The journal is now integrated with Wiley's Image Checking service. For further details,

see: <https://www.wiley.com/en-us/network/publishing/research-publishing/trending-stories/upholding-image-integrity-wileys-image-screening-service>

We look forward to receiving your revised submission.

Yours sincerely,

Karyn Hamilton
Senior Editor
The Journal of Physiology

REQUIRED ITEMS

-The Reference List must be in Journal format

-Your manuscript must include a complete Additional Information section

-The Journal of Physiology funds authors of provisionally accepted papers to use the premium BioRender site to create high resolution schematic figures. Follow this link and enter your details and the manuscript number to create and download figures. Upload these as the figure files for your revised submission. If you choose not to take up this offer we require figures to be of similar quality and resolution. If you are opting out of this service to authors, state this in the Comments section on the Detailed Information page of the submission form. The link provided should only be used for the purposes of this submission. Authors will be charged for figures created on this premium BioRender account if they are not related to this manuscript submission.

-Please upload separate high-quality figure files via the submission form.

-Papers must comply with the Statistics Policy https://jp.msubmit.net/cgi-bin/main.plex?form_type=display_requirements#statistics

In summary:

-If $n \leq 30$, all data points must be plotted in the figure in a way that reveals their range and distribution. A bar graph with data points overlaid, a box and whisker plot or a violin plot (preferably with data points included) are acceptable formats.

-If $n > 30$, then the entire raw dataset must be made available either as supporting information, or hosted on a not-for-profit repository e.g. FigShare, with access details provided in the manuscript.

- n clearly defined (e.g. x cells from y slices in z animals) in the Methods. Authors should be mindful of pseudoreplication.

-All relevant n values must be clearly stated in the main text, figures and tables

-The most appropriate summary statistic (e.g. mean or median and standard deviation) must be used. Standard Error of the Mean (SEM) alone is not permitted, unless justified and presented alongside confidence intervals.

-Exact p values must be stated. Authors must not use 'greater than' or 'less than'. Exact p values must be stated to three significant figures even when 'no statistical significance' is claimed.

EDITOR COMMENTS

Reviewing Editor:

Thank you for submitting your manuscript to The Journal of Physiology to be considered for inclusion in the special issue on the physiology of ageing skeletal muscle and the protective effects of exercise. Two reviewers have assessed your manuscript and while both were complimentary on the quality of writing and potential impact of the work, they also raised several major concerns that need to be addressed. Specifically, there is skepticism that the quadriceps CSA measurements from MRI were potentially confounded by muscle damage/inflammation skewing the results, which is in part supported by the discrepancies in the findings from the MRI versus the fiber CSA data. The paper would also be strengthened by inclusion of whole muscle function data and/or a more comprehensive evaluation of protein content and phosphorylation states of key signaling pathways rather than a select few transcripts. The authors need to carefully evaluate their statistical approach and reporting of the statistics and results as there were several instances of confusion.

Senior Editor:

Thank you for submitting your original research manuscript to The Journal of Physiology in response to the Special Issue call for papers on the physiology of ageing skeletal muscle and the protective effects of exercise. We recruited two reviewers with expertise in areas closely related to your field of research to provide peer feedback. Both were complimentary on the quality of writing and potential impact of the work. However, both referees also had some major concerns that will require close attention. For example, as summarized by the Reviewing Editor, there is some concern that the MRI quadriceps CSA measurements were potentially confounded by muscle damage. The reviewers also commented that the study would be more impactful with inclusion of whole muscle function data and/or a more comprehensive evaluation of key signaling pathways. Careful evaluation of the statistical approach and reporting is also required. We would like to invite you to carefully address all of the reviewer concerns and revise your manuscript for reconsideration.

Please pay close attention to guidelines requiring that precise p values be given, rather than, for example, $p < 0.05$.

REFEREE COMMENTS

Referee #1:

This study examined the effect of a single, single leg, preabilitative resistance training session in healthy older men on cumulative and acute post-prandial myofibrillar protein synthesis (MyoPS) and muscle cross-sectional area (CSA) after 5 days of strict bed rest. Muscle CSA was reduced by bed rest in the non-exercise control leg at 40, 60 and 80% of muscle length and this was attenuated by the prior exercise bout at 40 and 60% of muscle length. Integrated MyoPS decreased with bed rest and this was attenuated by the exercise bout. Finally, acute post-prandial MyoPS was increased in response to ingestion of milk protein in the pre-exercise, but not control, leg. This work suggests that a single bout of resistance exercise may be an effective prehabilitation strategy to attenuate bed rest-induced decreases in MyoPS and muscle mass in older males.

The manuscript is generally well written. The data is novel, although a little limited without data on muscle fiber size, phospho signalling proteins, markers of muscle damage or pre/post bed-rest muscle function.

Specific Comments:

Page 8, para 1, line 9 - would be helpful to the reader to indicate why enoxaparin was administered.

Page 8, para 2, line 13 - can you provide an analysis of the amino acid composition of the protein isolate?

Page 10, para 2, line 7 - what is the basis for this 6% decay? Can you provide a reference for the reader?

Page 10, para 2 - please provide the intra-individual day-to-day variability for these muscle CSA measures, specifically in your hands?

Is it at all possible that the attenuation of the decrease in muscle CSA is, in part, due to swelling associated with muscle damage induced by the bout the resistance exercise? A relatively high-volume training session in individuals that have not resistance training for at least ten years (or ever?) would surely lead to muscle damage with the potential for some oedema. Indeed, this is suggested as a possibility in the 2021 paper (Smeuninx, JCSM), which uses the same exercise protocol, and is supported by the study by Damas et al (2016, J Physiol) which used an exercise protocol of lower volume and similar intensity. There should be more discussion on the possibility.

Page 15, para 3, line 2 - change "or" for "for"

Page 16, para 2, line 6 - is the 40% Con value correct? Based on the mean values in Table 3, I calculate a different value (~2.3%). Please check.

Page 17, para 1, line 2 - what is the methodological significance of this change in body water enrichment?

Page 17, para 2, line 1-2 - is the P value for time correct?

Page 18, para 2, lines 7-8 - what is the rationale for analysing the changes in the abundance of two mRNAs that encode for proteins (p70S6K1 and mTOR) involved in a signaling pathway that is regulated by changes in protein phosphorylation? What would a change in either of these mRNAs mean for the regulation of this pathway, protein synthesis and/or muscle mass? Given that these are not referred to in the Discussion, I would recommend omitting these. A more appropriate measure would be to analyse the protein content of these targets and their phosphorylation status.

Related to this point, the final paragraph of the Introduction state, "The expression of key regulatory signaling proteins was also investigated", however, there is no protein data in this manuscript. Furthermore, the hypothesis also mentions that there would be preserved anabolic signalling expression. The wording should be amended to be more precise.

Page 18, para 2 and Figure 6 - the main text mentions Figure 5, however, the corresponding figure is Figure 6. Furthermore, some of the details in the legend of Figure 6 do not match the data in the figure.

Would the resistance exercise-attenuation in muscle CSA have any impact of muscle function? Ultimately the goal would be to prevent a decline in muscle function due to prolonged inactivity to allow a fast return to normal activities of daily living and prevent a transient acceleration in the age-related decline in strength.

Referee #2:

The present paper provides interesting data for the field and builds on the authors' previous report in JCSM. The paper is generally well written and the application of gold standard measures such as MRI and stable isotope tracers is a major strength. I am enthused by the manuscript as I believe it has the potential to be highly influential, but I do have several comments/sticking points that need to be addressed.

Major

1. The clinical trials registration indicates two primary outcomes. Moreover, the post bedrest acute FSR measures are not reported as a secondary outcome. It seems as if this pre-registration covers both the present and JCSM project but if there are two primary outcomes typically the p value for those outcomes should be halved to minimize the chance of a type 1 error, but that doesn't seem to be the case in the present report. Can the authors justify their approach? (Clinical trials is spelled incorrectly in the MS).

2. I also note that one primary outcome is stated as muscle mass/volume yet quadriceps at various CSAs is used in the study and volume is not reported?

3. Like my point above, you state that you are going to perform interactions and only perform post hocs when interactions are significant. However, it appears as if you have performed post hocs despite non-sig interactions (e.g. Figure 4). I encourage the authors to clearly report p value from interaction first for each outcome. From there, as you state in your own stats section, should not be doing post hocs that look across time within a given group. There appears to be some inconsistencies in the statistical reporting as well. 'A significant main effect for time (P= 0.311) was observed, but no interaction effect was observed'. But the P value is 0.311. Please address.

4. No power calculation was reported, and it seems like some of the outcomes are underpowered. What was the justification for the sample size?

5. Perhaps my primary comment relates to the timing of the MRI that was a day after the bout of RE and hours after a biopsy. It seems plausible that the RE bout may have induced SKMM damage leading to artificially increased quad volume. Careful inspection of the pre quad CSA data suggest that the EX leg volume was numerically higher at pre vs control adding weight to this concern. The authors even acknowledge that the damage may have been low to moderate. Was there baseline differences in Quad CSA in the ex leg? Can you provide sample MRI images and/or data related to circulating markers of damage? I appreciate circulating markers aren't perfect, but I encourage the authors to address this issue one way or another. Or I could just be wrong.

6. The authors make several statements like those listed below that read as if the bout of RE attenuated the decline in MPS and atrophy in the CTRL leg. Perhaps it's just me but I would suggest rephrasing as I assume the authors are not suggesting that there is a systemic anabolic effect of the RE bout in one leg transferring to the CTRL?

Page 2 Key Points

"... we demonstrate that a single bout of unilateral resistance exercise, performed the evening prior to 5-days of bed-rest, attenuated the decline in myofibrillar protein synthesis and muscle atrophy seen in the non-exercised leg in older men."

Page 18 Discussion

"The present study demonstrates that a single bout of unilateral leg resistance exercise (EX) performed the evening prior to 5-days of bed-rest, attenuated the rate of decline in iMyoPS and quadriceps CSA observed in the non-exercised leg (CTL) of older individuals"

Page 22 Discussion

"However, a single bout of RE performed on the evening prior to bed-rest attenuated the decline in quadriceps muscle mass observed in CTL at 40% (0.43% vs. 1.12% respectively, for EX and CTL) and 60% of muscle length (1.85 vs. 3.28% for EX and CTL, respectively).

7. Was the Ex leg randomly assigned?

Minor

8. You state that the single biopsy approach was used to assess acute MPS and cite Burd et al. However, in your methods you state that plasma $^{13}\text{C}_6$ phe enrichment was used as a proxy for muscle. But shouldn't it be plasma protein $^{13}\text{C}_6$ phe enrichment? Please clarify.

9. Some definitions are redefined twice (iMyoPS).

10. I don't believe the citations support the statement in the introduction that MyoPS is elevated 72h post exercise. The Burd paper is 48h?

11. Define LoS

12. Was the subcutaneous 20 mg injection of enoxaparin injected in one go?

13. Choose vs chose

14. The opening paragraph of the discussion seems unnecessarily verbose. Perhaps streamline.

END OF COMMENTS

Confidential Review

14-Jul-2023

30th August 2023

Journal of Physiology

Re: Submission of revised manuscript. JP-RP-2023-285130 - *A single bout of prior resistance exercise attenuates declines in myofibrillar protein synthesis and muscle atrophy during bed-rest in older men.*

To the Senior Editor

Firstly, we would like to thank you for the opportunity to revise our manuscript and submit the amended version for consideration. Secondly, we wish to thank the reviewers for their constructive and insightful comments.

We have extensively revised the manuscript based on the comments of the reviewers. The specific changes are highlighted within the revised manuscript. Attached below is a detailed response to each reviewers' comments.

We hope that editor and reviewers appreciate the extent to which we have gone to amend the manuscript and agree that these amendments have strengthened the manuscript to a level suitable for publication in the Journal of Physiology. If there is anything further required, please do not hesitate to contact me.

Sincerely,

Leigh Breen, PhD
Professor of Translational Muscle Physiology
MRC-ARUK Centre for Musculoskeletal Ageing Research
School of Sport, Exercise & Rehabilitation Sciences
University of Birmingham
Edgbaston
B15 2TT
United Kingdom

Tel: +44(0) 7885435500
Email: L.breen@bham.ac.uk

Reviewing Editor

Thank you for submitting your manuscript to The Journal of Physiology to be considered for inclusion in the special issue on the physiology of ageing skeletal muscle and the protective effects of exercise. Two reviewers have assessed your manuscript and while both were complimentary on the quality of writing and potential impact of the work, they also raised several major concerns that need to be addressed. Specifically, there is scepticism that the quadriceps CSA measurements from MRI were potentially confounded by muscle damage/inflammation skewing the results, which is in part supported by the discrepancies in the findings from the MRI versus the fiber CSA data. The paper would also be strengthened by inclusion of whole muscle function data and/or a more comprehensive evaluation of protein content and phosphorylation states of key signaling pathways rather than a select few transcripts. The authors need to carefully evaluate their statistical approach and reporting of the statistics and results as there were several instances of confusion.

Senior Editor

Thank you for submitting your original research manuscript to The Journal of Physiology in response to the Special Issue call for papers on the physiology of ageing skeletal muscle and the protective effects of exercise. We recruited two reviewers with expertise in areas closely related to your field of research to provide peer feedback. Both were complimentary on the quality of writing and potential impact of the work. However, both referees also had some major concerns that will require close attention. For example, as summarized by the Reviewing Editor, there is some concern that the MRI quadriceps CSA measurements were potentially confounded by muscle damage. The reviewers also commented that the study would be more impactful with inclusion of whole muscle function data and/or a more comprehensive evaluation of key signaling pathways. Careful evaluation of the statistical approach and reporting is also required. We would like to invite you to carefully address all of the reviewer concerns and revise your manuscript for reconsideration.

Please pay close attention to guidelines requiring that precise p values be given, rather than, for example, $p < 0.05$.

Thank-you to the Reviewing and Senior Editors for their comments. As you will see, we have provided detailed responses to hopefully satisfy concerns about the impact of exercise-induced inflammation/oedema on the MRI quadriceps CSA data. We have also added statistical data on fibre-type CSA comparisons between legs and explained the likely cause for the discrepancy between MRI and fibre-type CSA outcomes. Our responses to these concerns have been incorporated into the revised manuscript. We have justified aspects of our statistical approach. In acknowledgment of reviewer concerns and some errors in the original version, we have amended aspects of our statistics, results, and discussion sections to clarify our findings. Finally, we have performed additional analysis of total and phosphorylated signaling protein expression, as well as proxy blood markers of muscle damage, and incorporated these outcomes into the revised manuscript. Unfortunately, the invasive nature of the study impacted our ability to reliably measure muscle strength and function, as older participants could not exert close to a maximal effort in testing (we did attempt this with initial participants). Due to the influence of high-effort exercise on muscle protein turnover, conducting strength/function measures at any other time in the intervention would have confounded other outcomes. We have clarified this point in

the discussion and emphasized the importance of investigating the functional impact of acute RE prior to disuse, particularly in patient populations.

Reviewer 1:

This study examined the effect of a single, single leg, prehabilitative resistance training session in healthy older men on cumulative and acute post-prandial myofibrillar protein synthesis (MyoPS) and muscle cross-sectional area (CSA) after 5 days of strict bed rest. Muscle CSA was reduced by bed rest in the non-exercise control leg at 40, 60 and 80% of muscle length and this was attenuated by the prior exercise bout at 40 and 60% of muscle length. Integrated MyoPS decreased with bed rest, and this was attenuated by the exercise bout. Finally, acute post-prandial MyoPS was increased in response to ingestion of milk protein in the pre-exercise, but not control, leg. This work suggests that a single bout of resistance exercise may be an effective prehabilitation strategy to attenuate bed rest-induced decreases in MyoPS and muscle mass in older males.

The manuscript is generally well written. The data is novel, although a little limited without data on muscle fiber size, phospho signalling proteins, markers of muscle damage or pre/post bed-rest muscle function.

We thank reviewer 1 for these encouraging comments. We hope the responses below address the issues that have been highlighted. One point mentioned here in summary, but not in the specific comments below, is the absence of muscle fibre size data. However, muscle fibre CSA data were included in the reviewed manuscript. We have also discussed the fibre-type CSA findings in a response to a comment from reviewer 2, below.

Specific Comments:

Page 8, para 1, line 9 - would be helpful to the reader to indicate why enoxaparin was administered. **This information has been added.**

Page 8, para 2, line 13 - can you provide an analysis of the amino acid composition of the protein isolate? **We have added a reference to our previous work detailing the amino acid composition of the milk protein isolate.**

Page 10, para 2, line 7 - what is the basis for this 6% decay? Can you provide a reference for the reader? **Thank-you for pointing out this omission. After a single bolus of D2O we have previously seen deuterium loss at rates of ~6%/day (PMID: 24381002), therefore providing a daily top up of D2O that replenishes the lost 6%/day is required to maintain a constant steady state enrichment. We have added this information to the relevant section.**

Figure 3C shows that body water enrichment remained fairly constant with a slight decline overall. Given the inherent variability between individuals, the only way to keep a truly accurate steady state enrichment would be to measure each individual's personal decay rate and calculate a top-up dose based on that, which wasn't feasible for this study.

Page 10, para 2 - please provide the intra-individual day-to-day variability for these muscle CSA measures, specifically in your hands.

Intra-individual reliability for quadriceps CSA calculated from the repeated analysis of three MRI scans at 20, 40, 60 and 80% of limb length was 1.4, 1.0, 0.8 and 0.9%, respectively. We also observed a within-participant coefficient of variation for repeat day-to-day quadriceps CSA measurements to be 1.1%. This information has been added to the manuscript.

Is it at all possible that the attenuation of the decrease in muscle CSA is, in part, due to swelling associated with muscle damage induced by the bout the resistance exercise? A relatively high-volume training session in individuals that have not resistance training for at least ten years (or ever?) would surely lead to muscle damage with the potential for some oedema. Indeed, this is suggested as a possibility in the 2021 paper (Smeuninx, JCSM), which uses the same exercise protocol, and is supported by the study by Damas et al (2016, J Physiol) which used an exercise protocol of lower volume and similar intensity. There should be more discussion on the possibility. This is an interesting observation, and we have expanded the discussion to address this point. It is challenging to establish a RE protocol that would robustly stimulate MyoPS in older adults whilst avoiding any symptoms of muscle damage. However, we contend that the RE bout did not cause excessive damage that would explain the attenuated rate of quadriceps CSA loss in EX, based on the following points:

1. In our previous study (PMID: 33347733), we speculated that the iMyoPS response over bed-rest may have been directed towards the repair of damaged proteins caused by four bouts of RE prehabilitation performed over 7 days prior. In contrast, the current study used a single identical bout of prior RE the evening before bed-rest (25% of the load-stimulus in our previous study). Hence, we expect that the cumulative effects of 4 RE bouts in 7 days would elicit some degree of ‘damage’ symptoms, of greater magnitude than would be expected with a single bout of RE. Damas et al. (PMID: 27219125) demonstrated elevated quadriceps z-band streaming and circulating proxy markers of muscle ‘damage’ at 48h after an initial bout of RE (6 sets) performed to failure. The magnitude of change in these indirect markers of damage was interpreted to have caused ‘mild-to-moderate’ damage that may not necessarily have resulted in local inflammation and oedema. Indeed, in a separate study from the same project, Damas et al. (PMID: 26280652) reported relatively low-level muscle damage (ultrasound oedema, myoglobin, IL-6) after the initial bout of RE, whereas greater damage was apparent in the third week of training after 6 RE sessions had been completed.

2. Swelling/oedema, measured by ultrasound muscle thickness or echo intensity of the quadriceps, typically peaks in the early hours of RE recovery, and normally returned to pre-exercise levels 48 post-exercise (e.g., PMID: 37398959, PMID: 21411367, PMID: 34192417). The time-course of muscle oedema may extend further in response to highly damaging eccentric contractions. There is currently no evidence that older adults experience greater symptoms of exercise-induced muscle damage (PMID: 37395837). However, whilst the present RE was not overtly damaging as high eccentric-only protocol, we cannot rule out that some minor/transient soreness, reduced function/ROM, and possible oedema could have occurred. We used ultrasound to measure muscle thickness of the *vastus lateralis*, and summed *vastus lateralis*, *vastus intermedialis*, and *rectus femoris* thickness at 50% of limb length prior to RE, 18h post-RE (the onset of bed-rest on day 8, several hours prior to the

baseline MRI scan) and at the end of bed-rest in a sub-set of 8/10 participants. Images were obtained longitudinally to the thigh along the sagittal axis of the muscle, with an intra-operator reliability of 1.4% for three images. As shown below, we did not observe any significant difference in muscle thickness from pre-RE values in EX or CTL, suggesting that any oedema was not present/detectable or had subsided and would not influence quadriceps CSA measures. We could not reliably measure ultrasound echo intensity to corroborate the muscle thickness findings. Furthermore, we were unable to determine temporal changes in proxy circulating markers of damage during bed-rest as samples were obtained solely for clinical analysis of coagulation markers. Although not as insightful as the initial days of post-exercise recovery, we have now analysed plasma concentrations of creatine kinase and myoglobin, obtained during preliminary testing and the day after bed-rest. These data are presented in the revised manuscript.

The fact we did not detect any effect of bed-rest on ultrasound-based muscle thickness (e.g., to indicate muscle atrophy), or a difference between CTL and EX, is most likely explained by our use of one-dimensional ultrasound muscle thickness measurements. Specifically, others have shown a relationship between MRI-and ultrasound-based quadriceps disuse atrophy when serial image stitching was used to quantify quadriceps CSA (PMID: 33403796). In contrast, the use of one-dimensional muscle thickness measurements to assess muscle size is thought to be useful at a single-time point but may offer little insight into changes in muscle size over time (PMID: 28805932).

3. It was recently reported that 300 eccentric contractions (to induce severe muscle damage of the quadriceps) immediately before limb immobilization, transiently prevented early quadriceps muscle atrophy when examining absolute changes in MRI-derived muscle volume (PMID: 34632796). This apparent protective effect of exercise remained when the authors corrected between immobilized and non-immobilized legs to account for damaged-induced oedematous swelling of both legs and changes in volume to the non-immobilized control leg. These data suggest that any difference in quadriceps volume between legs had to be from disuse.

4. We provide mechanistic evidence that the protective effect of prior RE were driven by changes in myofibrillar protein synthesis and not muscle damage/oedema per se. We report an association between the decline in iMyoPS and quadriceps CSA in CTL, that was not present in EX, suggesting that MyoPS rates were likely maintained for longer during bed-rest by the prior RE stimulus.

We have provided a concise summary of the above points in the relevant section (paragraph 2 of the discussion). We have opted not to include a detailed description of the ultrasound methodology and results, but instead make reference to these findings in the discussion. This decision was taken to avoid opening up lengthy discussion sections related to the issues outlined above, that detract from the narrative of the manuscript. We hope this approach is agreeable to the reviewer.

Page 15, para 3, line 2 - change "or" for "for" **Corrected**

Page 16, para 2, line 6 - is the 40% Con value, correct? Based on the mean values in Table 3, I calculate a different value (~2.3%). Please check. **Thank you for checking and highlighting this error, the correct decline in CTL is 1.12%. This has now been corrected in table 3.**

Page 17, para 1, line 2 - what is the methodological significance of this change in body water enrichment? **Thank you for this observation, which is an important point to clarify. The direct incorporation of stable isotope tracers into new muscle proteins is considered the most ideal approach to determine rates of muscle protein synthesis. The measured bound enrichment in muscle relies on the both the rate of incorporation and the precursor enrichment of the body water. As such, when muscle protein synthesis is calculated, the slight drop of enrichment is accounted for in the precursor-product calculation, with protein synthesis calculations possible in steady and non-steady state conditions (PMID: 16368786). For instance, we have previously demonstrated constants rates of muscle protein synthesis over 8 days using a single bolus of D2O where body water enrichment declines over that time (PMID: 24381002). Similarly, others investigating unilateral immobilisation have shown an overall increase in body water enrichment over the study period (PMID: 34632796, PMID: 31743039), with a constant rate of MPS in the control legs, and an overall decline in muscle protein synthesis in the immobilised legs. Our aim was to maintain body water enrichments over the study, however the small decline in enrichment will not have impacted calculated rates of MPS. In addition, a strength of this study is the unilateral design where both legs experience the same period of body water enrichment over the study period. We have added some additional text to the results section to add clarity to this point.**

Page 17, para 2, line 1-2 - is the P value for time, correct? **This p value was incorrect and has been amended to P=0.0311.**

Page 18, para 2, lines 7-8 - what is the rationale for analysing the changes in the abundance of two mRNAs that encode for proteins (p70S6K1 and mTOR) involved in a signaling pathway that is regulated by changes in protein phosphorylation? What would a change in either of these mRNAs mean for the regulation of this pathway, protein synthesis and/or muscle mass? Given that these are not referred to in the Discussion, I would recommend omitting these. A more appropriate measure would be to analyse the protein content of these targets and their phosphorylation status. **Thank you for highlighting this point. Gene expression was included as a read-out for the capacity for functional protein production. We appreciate that alterations in intramuscular gene expression are generally transient events, but others have previously shown a change in anabolic (PMID 22338078) and catabolic (PMID 26578714, PMID: 26173027) gene expression with bed-rest and immobilization. Nonetheless, we agree that changes in total and phosphorylated protein represent more viable read-out for alterations in muscle proteostasis. We have therefore analysed total and phosphorylated protein expression of intermediates involved in postprandial aMyoPS stimulation via immunoblot and added these outcomes to the manuscript. These signaling targets were selected based on our interpretation that greater iMyoPS over bed-rest in EX may have been underscored by preserved aMyoPS stimulation. Unfortunately, we were unable to reliably analyse the total protein content of myostatin and other markers of proteolysis that were presented for gene expression (e.g., non-specific bands or no bands present at the expected MW). In any case, there was no suggestion from the gene expression data that these markers of proteolysis differed between EX and CTL.**

Related to this point, the final paragraph of the Introduction state, "The expression of key regulatory signaling proteins was also investigated", however, there is no protein data in this manuscript. Furthermore, the hypothesis also mentions that there would be preserved anabolic signalling expression. The wording should be amended to be more precise. **As mentioned above, we have now added protein phosphorylation targets to the manuscript and amended the introduction to refer to these specific outputs.**

Page 18, para 2 and Figure 6 - the main text mentions Figure 5, however, the corresponding figure is Figure 6. Furthermore, some of the details in the legend of Figure 6 do not match the data in the figure. **Thank you for highlighting these errors. We have amended the results text figure reference and corrected the details in figure 6 legend.**

Would the resistance exercise-attenuation in muscle CSA have any impact of muscle function? Ultimately the goal would be to prevent a decline in muscle function due to prolonged inactivity to allow a fast return to normal activities of daily living and prevent a transient acceleration in the age-related decline in strength. **Thank you for this insightful comment. We fully agree that the impact of the current exercise intervention on function/strength is of great relevance to numerous patient outcomes (e.g., return to normal IADL, risk of fracture, falls, readmission, external care). Our intention was to measure maximal voluntary and twitch interpolated strength of the knee extensors prior to and following bed-rest via dynamometry. Based on previous**

findings (PMID: 28482746) we would have expected to see a decline in maximal strength with bed-rest. Due to logistical challenges of the post-bed rest experimental trial, the assessments of strength had to be conducted at the end of the stable isotope infusion and serial biopsy procedures (to avoid an acute effect of contraction on aMyoPS). Initial study participants indicated that they were unable to give a maximal effort to strength testing due to post-biopsy tenderness in the quadriceps. Hence, we decided to omit function/strength testing assessments from the testing battery for all subsequent participants. We have attempted, throughout the discussion, to acknowledge that the functional relevance of the single-bout resistance exercise protocol is an essential consideration for future studies, particularly in a clinical context.

Reviewer 2:

The present paper provides interesting data for the field and builds on the authors' previous report in JCSM. The paper is generally well written and the application of gold standard measures such as MRI and stable isotope tracers is a major strength. I am enthused by the manuscript as I believe it has the potential to be highly influential, but I do have several comments/sticking points that need to be addressed. **We thank reviewer 2 for the positive comments regarding the quality of writing, methodological strengths, and the potential influence of our work.**

1. The clinical trials registration indicates two primary outcomes. Moreover, the post bedrest acute FSR measures are not reported as a secondary outcome. It seems as if this pre-registration covers both the present and JCSM project but if there are two primary outcomes typically the p value for those outcomes should be halved to minimize the chance of a type 1 error, but that doesn't seem to be the case in the present report. Can the authors justify their approach? (Clinical trials is spelled incorrectly in the MS). **We thank the reviewers for highlighting this. The original project proposal was designed with (and powered to) a single primary outcome of muscle mass change over bed-rest in untrained control legs, or a leg that had undergone prior single or multiple exercise bouts. The change in integrated and acute myofibrillar FSR belongs as a secondary outcome measure and has now been corrected and were approved by our local governance.**

2. I also note that one primary outcome is stated as muscle mass/volume yet quadriceps at various CSAs is used in the study and volume is not reported? **During pilot scanning we were unable to obtain sufficient images across the length of the thigh in the designated time, which would have allowed us to reliably quantify overall quadriceps volume. Given evidence that quadriceps atrophy during disuse may differ across regions, the decision was made to image proximal-to-distal CSAs to capture changes across limb length. This updated information has now been included in our trial registry.**

3. Like my point above, you state that you are going to perform interactions and only perform post hocs when interactions are significant. However, it appears as if you have performed post hocs despite non-sig interactions (e.g., Figure 4). I encourage the authors to clearly report p value from interaction first for each outcome. From there, as you state in your own stats section, should not be doing post hocs that look across time within a given group. There appears to be some inconsistencies in the statistical

reporting as well. 'A significant main effect for time ($P=0.311$) was observed, but no interaction effect was observed'. But the P value is 0.311. Please address.

The incorrect p value of 0.311 was a typo, which has now been amended in the manuscript to 0.0311.

Regarding the reporting of post-hoc multiple comparisons in the absence of interaction effects, we have amended the statistics section to reflect our decision to report post-hocs when main or interaction effects were evident. Post-hoc analysis of factor effects is generally based on treatment means (for multiple comparisons) when the interaction effect is statistically significant. However, the idea that comparisons among treatment means should not be made when an interaction effect is not significant, has been questioned (PMID: 21547361). Comparisons among the treatment means can be performed when there is a non-significant interaction effect, as a way to further the analysis of experimental data and draw potentially meaningful conclusions. In this scenario, performing specific post-hoc testing is commonly performed in the field (e.g., PMID: 37598750, PMID: 30694972, PMID: 33601335) and statistically justified, as these tests are data-driven (a posteriori) and more conservative. In line with the reviewer's recommendation, we have reported the interaction effect first for all outcomes and have referred to the lack of interaction effect when interpreting the findings in the discussion.

4. No power calculation was reported, and it seems like some of the outcomes are underpowered. What was the justification for the sample size? Thank-you for highlighting this omission. We have added details of the sample size calculations to the statistics section of the methods. Secondary mechanistic outcomes were considered supportive or explanatory for the primary outcome and, as such, were not accounted for in the sample size calculations. We acknowledge that some outcomes (e.g., iMyoPS and aMyoPS) may be moderately underpowered to detect significant interaction effects that would strengthen our interpretation of the findings. However, it was not possible to continue recruiting due to high cost (>£5000 p/p through testing only) and time constraints. We have further acknowledged the issue of sample size and statistical power in the discussion.

5. Perhaps my primary comment relates to the timing of the MRI that was a day after the bout of RE and hours after a biopsy. It seems plausible that the RE bout may have induced SKMM damage leading to artificially increased quad volume. Careful inspection of the pre quad CSA data suggest that the EX leg volume was numerically higher at pre vs control adding weight to this concern. The authors even acknowledge that the damage may have been low to moderate. Were there baseline differences in Quad CSA in the ex leg? Can you provide sample MRI images and/or data related to circulating markers of damage? I appreciate circulating markers aren't perfect, but I encourage the authors to address this issue one way or another. Or I could just be wrong. Thank-you for raising this important point, which relates to similar concerns from reviewer 1, to which we have responded in detail. As you will see from the above response, we are confident that prior RE did not artificially increase quadriceps CSA, at least not in the regions where rates of subsequent loss and between limb differences were most apparent. Comparison of pre bed-rest quadriceps CSA at 20, 40, 60 and 80% revealed no statistically significant differences between legs ($P=0.13$, 0.30 , 0.37 and 0.42 , respectively). The numerically greater quadriceps CSA values in EX may be explained by the selection of the strongest/dominant leg to undertake the exercise (as described in the methods). Similarly, the pattern of response showing

an increase in fibre CSA in EX and a decrease in fibre CSA for CTL can likely be explained the inherent inter- and intra-individual variability in fibre-type size when sampling from different longitudinal sites (PMID: 34013752) in a sample of this size. The delta changes in fibre-type CSA from pre-to-post bed-rest did not differ between EX and CTL ($P=0.337$ and 0.422 for type I and II, respectively), which has now been included in the results. Hence, we do not think the observed responses is evidence of oedema that would explain the protective effects of prior RE on quadriceps CSA. We also believe that the variability issue described above can partly explain the discrepancy between the findings of MRI-based quadriceps CSA and fibre-type CSA data.

Regarding the influence of prior biopsy sampling causing a damage response that would influence MRI-based quadriceps CSA, we acknowledge that the scanning timing was not optimal but was necessary given the logistical challenges of our trial. Post-biopsy swelling to a level that would be detectable on an MRI scan is rare and would result only from intramuscular haematoma. Such incidents are quite rare (~1:250) but never mild and easily visible to investigators and nursing staff (images can be provided if requested). Our standard practice is to apply direct pressure to the biopsy site with sterile gauze and an ice pack for 10 minutes afterwards to minimize potential inflammation and swelling, and ensure haemostasis is reached before wound closure. MRI scans were performed after the same number of biopsies in CTL and EX legs (e.g., after 1 biopsy per leg on Day 8 and 2 biopsies per leg on Day 13). Any incidence of haematoma was a contraindication to further study procedures, including MRI scanning.

As you will also see in our response to reviewer 1, blood samples obtained over bed-rest in the days following RE were solely for rapid clinical analysis of coagulation markers for safety purposes. Therefore, we are unable to track changes in circulating proxy markers of damage in response to single leg RE that would usually peak between 24-72 post-exercise. The only remaining available blood samples were obtained in the fasted state during preliminary testing and the day after bed-rest (6-days post-RE). We have analysed plasma concentrations of creatine kinase and myoglobin concentrations in these samples, showing no difference between time-points. Data are reported in Tale 4 of the manuscript.

6. The authors make several statements like those listed below that read as if the bout of RE attenuated the decline in MPS and atrophy in the CTRL leg. Perhaps it's just me but I would suggest rephrasing as I assume the authors are not suggesting that there is a systemic anabolic effect of the RE bout in one leg transferring to the CTRL? Thank you for highlighting the lack of clarity in the writing here. No, we are not suggesting any systemic effect that would protect muscle mass or iMyoPS in the CTRL leg, but are attempting to highlight that the observed decline in CTL was mitigated in EX. To clarify this point, we have addressed the specific sentences highlighted below, and others in the manuscript.

Page 2 Key Points

"... we demonstrate that a single bout of unilateral resistance exercise, performed the evening prior to 5-days of bed-rest, attenuated the decline in myofibrillar protein synthesis and muscle atrophy seen in the non-exercised leg in older men." **Re-written.**

Page 18 Discussion

"The present study demonstrates that a single bout of unilateral leg resistance exercise (EX) performed the evening prior to 5-days of bed-rest, attenuated the rate of decline in iMyoPS and quadriceps CSA observed in the non-exercised leg (CTL) of older individuals" **Re-written.**

Page 22 Discussion

"However, a single bout of RE performed on the evening prior to bed-rest attenuated the decline in quadriceps muscle mass observed in CTL at 40% (0.43% vs. 1.12% respectively, for EX and CTL) and 60% of muscle length (1.85 vs. 3.28% for EX and CTL, respectively). **Re-written.**

7. Was the Ex leg randomly assigned? **No. we chose to perform the exercise on the strongest leg, as determined by 1RM testing during preliminary testing visits. This information is provided on page 7.**
8. You state that the single biopsy approach was used to assess acute MPS and cite Burd et al. However, in your methods you state that plasma ¹³C₆ phe enrichment was used as a proxy for muscle. But shouldn't it be plasma protein ¹³C₆ phe enrichment? Please clarify. **We have amended the text to reflect that this was a measure of mixed plasma protein ¹³C₆ phenylalanine enrichment.**
9. Some definitions are redefined twice (iMyoPS). **Amended throughout.**
10. I don't believe the citations support the statement in the introduction that MyoPS is elevated 72h post exercise. The Burd paper is 48h? **The citations listed show that MyoPS is elevated at 24 h (Burd), 48 h (McKendry) and 72 h (Miller) post-exercise. Our point was to highlight the timeframe over which alterations in muscle protein turnover may be apparent. To avoid any confusion, we have altered the text slightly to reflect that MyoPS may remain elevated for up to 72h post-exercise.**
11. Define LoS **The abbreviation for length of stay has now been defined earlier in the manuscript.**
12. Was the subcutaneous 20 mg injection of enoxaparin injected in one go? **Yes, this was a single once daily single injection, at approximately the same time of day during bed-rest (typically after breakfast). We have added additional detail to the methods description.**
13. Choose vs chose **Amended.**
14. The opening paragraph of the discussion seems unnecessarily verbose. Perhaps streamline. **Thank-you for the suggestion, we have streamlined the opening discussion paragraph.**

Dear Professor Breen,

Re: JP-RP-2023-285130R1 "A single bout of prior resistance exercise attenuates muscle atrophy and declines in myofibrillar protein synthesis during bed-rest in older men." by Benoit Smeuninx, Yasir S Elhassan, Elizabeth Sapey, Paul T Morgan, Marie Korzepa, Archie E Belfield, Alison Rushton, Andrew Philp, Nima Gharahdaghi, Matthew S Brook, Daniel James Wilkinson, Ken Smith, Philip J Atherton, and Leigh Breen

Thank you for submitting your manuscript to The Journal of Physiology. It has been assessed by a Reviewing Editor and by 2 expert referees and we are pleased to tell you that it is acceptable for publication following satisfactory revision.

REVISION CHECKLIST:

Please upload two versions of your manuscript text: one with all relevant changes highlighted and one clean version with no changes tracked. The manuscript file should include all tables and figure legends, but each figure/graph should be uploaded as separate, high-resolution files. The journal is now integrated with Wiley's Image Checking service. For further details, see: <https://www.wiley.com/en-us/network/publishing/research-publishing/trending-stories/upholding-image-integrity-wileys-image-screening-service>.

We look forward to receiving your revised submission.

Yours sincerely,

Karyn Hamilton
Senior Editor
The Journal of Physiology

REQUIRED ITEMS

-Your manuscript must include a complete Additional Information section
Currently, the 'Author Contributions' section appears to be missing from the article?

-You must upload original, uncropped western blot/gel images (including controls) if they are not included in the manuscript. This is to confirm that no inappropriate, unethical or misleading image manipulation has occurred <https://physoc.onlinelibrary.wiley.com/hub/journal-policies#imagmanip> These should be uploaded as 'Supporting information for review process only'. Please label/highlight the original gels so that we can clearly see which sections/lanes have been used in the manuscript figures.

-Papers must comply with the Statistics Policy https://jp.msubmit.net/cgi-bin/main.plex?form_type=display_requirements#statistics

In summary:

-If $n \leq 30$, all data points must be plotted in the figure in a way that reveals their range and distribution. A bar graph with data points overlaid, a box and whisker plot or a violin plot (preferably with data points included) are acceptable formats.

-If $n > 30$, then the entire raw dataset must be made available either as supporting information, or hosted on a not-for-profit repository e.g. FigShare, with access details provided in the manuscript.

- n clearly defined (e.g. x cells from y slices in z animals) in the Methods. Authors should be mindful of pseudoreplication.

-All relevant n values must be clearly stated in the main text, figures and tables

-The most appropriate summary statistic (e.g. mean or median and standard deviation) must be used. Standard Error of the Mean (SEM) alone is not permitted, unless justified and presented alongside confidence intervals.

-Exact p values must be stated. Authors must not use 'greater than' or 'less than'. Exact p values must be stated to three significant figures even when 'no statistical significance' is claimed.

EDITOR COMMENTS

Reviewing Editor:

Thank you for submitting your work to The Journal of Physiology and for providing a thorough revision of your manuscript. All major concerns have been adequately addressed. However, the investigators did not provide the original, uncropped gels during the manuscript submission, which is a requirement by The Journal before a final decision on acceptance can be made.

Senior Editor:

Thank you for submitting your revised work to The Journal of Physiology. All major concerns have been adequately addressed. However, as noted by the Reviewing Editor, you still must provide the original, uncropped gels. I've also noted a number of inconsistencies with the Journal Statistics Policy that need to be addressed. Finally, please confirm any/which data presented here are published elsewhere along with a rationale for doing so.

The authors note that they present some data as mean \pm SD (which is consistent with Journal policy. However, they indicate that other data (morphology and biological sample data) are presented as mean \pm SEM (not consistent with Journal policy. The Journal Statistics policy also indicates that precise p-values be provided, not just $p < 0.01$, for example. The authors provide precise p-values in many cases, but in others, do not. Please address these inconsistencies. Thank you.

REFEREE COMMENTS

Referee #1:

The Authors are to be commended for their considered responses to my queries and for the amendments made to the manuscript. I have no further comments.

Referee #2:

Thank you for addressing my concerns. My primary issue was the timing of the MRI around the exercise protocol that has now been resolved.

END OF COMMENTS

1st Confidential Review

14-Sep-2023

Reviewing Editor

Thank you for submitting your work to The Journal of Physiology and for providing a thorough revision of your manuscript. All major concerns have been adequately addressed. However, the investigators did not provide the original, uncropped gels during the manuscript submission, which is a requirement by The Journal before a final decision on acceptance can be made.

Senior Editor

Thank you for submitting your revised work to The Journal of Physiology. All major concerns have been adequately addressed. However, as noted by the Reviewing Editor, you still must provide the original, uncropped gels. I've also noted a number of inconsistencies with the Journal Statistics Policy that need to be addressed. Finally, please confirm any/which data presented here are published elsewhere along with a rationale for doing so.

The authors note that they present some data as mean \pm SD (which is consistent with Journal policy). However, they indicate that other data (morphology and biological sample data) are presented as mean \pm SEM (not consistent with Journal policy). The Journal Statistics policy also indicates that precise p-values be provided, not just $p < 0.01$, for example. The authors provide precise p-values in many cases, but in others, do not. Please address these inconsistencies. Thank you.

To the Reviewing and Senior Editors, thank-you for these additional pointers on manuscript formatting. We have uploaded the uncropped western blot images with our revised submission. No data from the current manuscript have been published elsewhere. We have added a note to the methods section to confirm this. Finally, we have amended the results and methods to state SD rather than SEM, and accurate p values have been added throughout. Below, we have provided additional comments to other minor points that were raised.

REQUIRED ITEMS

-Your manuscript must include a complete additional information section. Currently, the 'Author Contributions' section appears to be missing from the article? **This issue has been rectified and an 'author contributions' statement added to the additional information.**

-You must upload original, uncropped western blot/gel images (including controls) if they are not included in the manuscript. This is to confirm that no inappropriate, unethical or misleading image manipulation has occurred. These should be uploaded as 'Supporting information for review process only'. Please label/highlight the original gels so that we can clearly see which sections/lanes have been used in the manuscript figures. **Uncropped western images have been uploaded with our resubmission.**

-Papers must comply with the Statistics Policy

In summary:

-If $n \leq 30$, all data points must be plotted in the figure in a way that reveals their range and distribution. A bar graph with data points overlaid, a box

and whisker plot or a violin plot (preferably with data points included) are acceptable formats. **Individual data points are plotted within figures to reveal the range and distribution of data.**

-If $n > 30$, then the entire raw dataset must be made available either as supporting information, or hosted on a not-for-profit repository e.g., FigShare, with access details provided in the manuscript. **N/A to this manuscript.**

-'n' clearly defined (e.g., x cells from y slices in z animals) in the Methods. Authors should be mindful of pseudoreplication. **All n values are correctly detailed in the methods.**

-All relevant 'n' values must be clearly stated in the main text, figures and tables **All n values are correctly stated in main text, figures and tables.**

-The most appropriate summary statistic (e.g. mean or median and standard deviation) must be used. Standard Error of the Mean (SEM) alone is not permitted, unless justified and presented alongside confidence intervals. **We have replaced any SEM data with SD and amended the statistics section and figure legends accordingly.**

-Exact p values must be stated. Authors must not use 'greater than' or 'less than'. Exact p values must be stated to three significant figures even when 'no statistical significance' is claimed. **Exact p values have now been added throughout the results section with the exception of $P < 0.001$. Non-significant p values have also been added, where appropriate.**

Dear Professor Breen,

Re: JP-RP-2023-285130R2 "A single bout of prior resistance exercise attenuates muscle atrophy and declines in myofibrillar protein synthesis during bed-rest in older men." by Benoit Smeuninx, Yasir S Elhassan, Elizabeth Sapey, Paul T Morgan, Marie Korzepa, Archie E Belfield, Alison Rushton, Andrew Philp, Nima Gharahdaghi, Matthew S Brook, Daniel James Wilkinson, Ken Smith, Philip J Atherton, and Leigh Breen

Thank you for submitting your manuscript to The Journal of Physiology. It has been assessed by a Reviewing Editor and we are pleased to tell you that it is acceptable for publication following satisfactory revision.

The editorial reports are copied at the end of this email.

REVISION CHECKLIST:

Please upload two versions of your manuscript text: one with all relevant changes highlighted and one clean version with no changes tracked. The manuscript file should include all tables and figure legends, but each figure/graph should be uploaded as separate, high-resolution files. The journal is now integrated with Wiley's Image Checking service. For further details, see: <https://www.wiley.com/en-us/network/publishing/research-publishing/trending-stories/upholding-image-integrity-wileys-image-screening-service>.

We look forward to receiving your revised submission.

Yours sincerely,

Karyn Hamilton
Senior Editor
The Journal of Physiology

EDITOR COMMENTS

Reviewing Editor:

Thank you for the prompt submission of your revised article to The Journal of Physiology. All revisions have been addressed other than the proper formatting of the whole, original uncropped gels. The details of The Journal's requirements can be found under The Journal's information for authors section (https://jp.msubmit.net/cgi-bin/main.plex?form_type=display_requirements#Revised%20submissions), but in brief, the gels must include the labelling of all lanes, including an appropriate loading control and the molecular weight markers, before a final decision on acceptance can be made.

Senior Editor:

Thank you for the revisions to bring your manuscript closer to being in line with Journal requirements. As pointed out by the Reviewing Editor, the blots must include the labeling of all lanes, including the corresponding kDa of the molecular weight marker bands, before a final decision on acceptance can be made. The images you provided do not appear to be full length; please provide an explanation for this OR the full length images. Finally, can you please clarify if and how you used the Ponceau as a loading control. Thank you for your attention to detail on this.

END OF COMMENTS

2nd Confidential Review

26-Sep-2023

Reviewing Editor:

Thank you for the prompt submission of your revised article to The Journal of Physiology. All revisions have been addressed other than the proper formatting of the whole, original uncropped gels. The details of The Journal's requirements can be found under The Journal's information for authors section (https://jp.msubmit.net/cgi-bin/main.plex?form_type=display_requirements#Revised%20submissions), but in brief, the gels must include the labelling of all lanes, including an appropriate loading control and the molecular weight markers, before a final decision on acceptance can be made.

Senior Editor:

Thank you for the revisions to bring your manuscript closer to being in line with Journal requirements. As pointed out by the Reviewing Editor, the blots must include the labelling of all lanes, including the corresponding kDa of the molecular weight marker bands, before a final decision on acceptance can be made. The images you provided do not appear to be full length; please provide an explanation for this OR the full length images. Finally, can you please clarify if and how you used the Ponceau as a loading control. Thank you for your attention to detail on this.

Response: We have updated the 'uncropped western images' attachment to include the molecular weight indicators and lane IDs on all images. We have added a note to this file to state that full-length uncropped images are not available as membranes were cut following ponceau staining to allow us to probe multiple targets on the same membrane for efficiency.

We have amended the immunoblot methods section to clarify exactly how ponceau was used as a loading control.

Dear Dr Breen,

Re: JP-RP-2023-285130R3 "A single bout of prior resistance exercise attenuates muscle atrophy and declines in myofibrillar protein synthesis during bed-rest in older men." by Benoit Smeuninx, Yasir S Elhassan, Elizabeth Sapey, Paul T Morgan, Marie Korzepa, Archie E Belfield, Alison Rushton, Andrew Philp, Nima Gharahdaghi, Matthew S Brook, Daniel James Wilkinson, Ken Smith, Philip J Atherton, and Leigh Breen

We are pleased to tell you that your paper has been accepted for publication in The Journal of Physiology.

Authors should note that it is too late at this point to offer corrections prior to proofing. The accepted version will be published online, ahead of the copy edited and typeset version being made available. Major corrections at proof stage, such as changes to figures, will be referred to the Editors for approval before they can be incorporated. Only minor changes, such as to style and consistency, should be made at proof stage. Changes that need to be made after proof stage will usually require a formal correction notice.

Yours sincerely,

Karyn Hamilton
Senior Editor
The Journal of Physiology

P.S. - You can help your research get the attention it deserves! Check out Wiley's free Promotion Guide for best-practice recommendations for promoting your work at www.wileyauthors.com/eeo/guide. You can learn more about Wiley Editing Services which offers professional video, design, and writing services to create shareable video abstracts, infographics, conference posters, lay summaries, and research news stories for your research at www.wileyauthors.com/eeo/promotion.

IMPORTANT NOTICE ABOUT OPEN ACCESS: To assist authors whose funding agencies mandate public access to published research findings sooner than 12 months after publication, The Journal of Physiology allows authors to pay an Open Access (OA) fee to have their papers made freely available immediately on publication.

You can check if your funder or institution has a Wiley Open Access Account here: <https://authorservices.wiley.com/author-resources/Journal-Authors/licensing-and-open-access/open-access/author-compliance-tool.html>.

EDITOR COMMENTS

Reviewing Editor:

Thank you for the prompt turnaround on the few minor revisions that remained from the last version. The revisions have been adequately addressed, and I would like to congratulate the authors on the completion of an excellent study. Thank you for submitting your work to the special issue on the Physiology of Ageing Skeletal Muscle and the Protective Effects of Exercise in The Journal of Physiology.

Senior Editor:

Thank you for quickly addressing those missing items!

3rd Confidential Review

03-Oct-2023